# Exo84c interacts with VAP27 to regulate exocytotic compartment degradation and stigma senescence

Tong Zhang[1,2], Yifan Li[1,2], Chengyang Li[1,2], Jingze Zang[1,2,3], Erlin Gao[1,2], Johan T. Kroon [3], Xiaolu Qu[1], Patrick J. Hussey [3] & Pengwei Wang [1,2] ✉

In plants, exocyst subunit isoforms exhibit significant functional diversity in that they are involved in either protein secretion or autophagy, both of which are essential for plant development and survival. Although the molecular basis of autophagy is widely reported, its contribution to plant reproduction is not very clear. Here, we have identified Exo84c, a higher plant-specific Exo84 isoform, as having a unique function in modulating exocytotic compartment degradation during stigmatic tissue senescence. This process is achieved through its interaction with the ER localised VAP27 proteins, which regulate the turnover of Exo84c through the autophagy pathway. VAP27 recruits Exo84c onto the ER membrane as well as numerous ER-derived autophagosomes that are labelled with ATG8. These Exo84c/exocyst and VAP27 positive structures are accumulated in the vacuole for degradation, and this process is partially perturbed in the *exo84c* knock-out mutants. Interestingly, the *exo84c* mutant showed a prolonged effective pollination period with higher seed sets, possibly because of the delayed stigmatic senescence when Exo84c regulated autophagy is blocked. In conclusion, our studies reveal a link between the exocyst complex and the ER network in regulating the degradation of exocytosis vesicles, a process that is essential for normal papilla cell senescence and flower receptivity.

Flowers are the most important organs for plants to produce progenies. The limited floral life span determines the effective pollination period (EPP)[1], which is known to be the key determinator of yield and is restricted by flower senescence. The floral organ senescence is triggered by successful pollination and senescence-induced programmed cell death (PCD) during regular development[2-5]. Most of the previous studies on flower senescence use petals as the model system and these studies have revealed important physiological and molecular mechanisms in this process[2,6,7], however, the correlation between floral receptivity and petal life span is rather indirect.

The stigma tissue is responsible for recognizing and accepting pollen and determines whether fertilization will take place[2,5,8,9]. A recent study shows that the senescence of the stigma is regulated by PCD which terminates floral receptivity in Arabidopsis flowers, and the seed-set rate is positively related to the viability of stigmatic papilla cells[5]. Autophagy is a precisely regulated mechanism composed of machinery that is involved in plant survival and senescence. Several autophagy-related genes have been identified with higher expression in stigmas that are undergoing senescence which suggests a requirement for autophagy in floral and stigma tissue viability[5,10]. However, the underlying cellular mechanism of the role that autophagy plays in

[1]National Key Laboratory for Germplasm Innovation & Utilization of Horticultural Crops, College of Horticulture and Forestry Sciences, Huazhong Agricultural University, Wuhan 430070 Hubei Province, China. [2]Hubei Hongshan Laboratory, Wuhan 430070, China. [3]Department of Biosciences, Durham University, South Road, Durham DH1 3LE, UK. ✉e-mail: wangpengwei@mail.hzau.edu.cn

regulating plant reproduction and floral senescence remains largely unknown. In addition to the core autophagy machinery (e.g. ATG genes), the ER network and ER intrinsic proteins are also essential in autophagosome biogenesis[11–16]. The ER/EPCS-resident protein, VAP/VAP27, is one of the best examples in eukaryotic cells. VAP/VAP27 interacts with multiple proteins that function in either the membrane trafficking or lipid transport pathways to regulate autophagosome biogenesis[17–22].

The exocyst complex is evolutionarily conserved, it is composed of eight subunits: Sec3, Sec5, Sec6, Sec8, Sec10, Sec15, Exo70 and Exo84. In yeast, all subunits can be co-purified and stay associated at the same time from a structural biology perspective[23]. In plants, most exocyst subunits have multiple isoforms that participate in exocytotic vesicle tethering, secretion[24,25] and autophagy[26]. The Exo70 family is the best example of such functional diversity[27]. Exo70A1 is required for compatible pollination in both *Brassica napus* and *Arabidopsis thaliana*[28,29]. Exo70B2 interacts directly with ATG8 and is transported into the vacuole through autophagy during immune responses[30]. Knocking out the Exo70D family results in plant death in carbon-deficient conditions[31]. Compared to the Exo70 proteins, most studies of the Exo84 family have been focused on the Exo84b isoform which is believed to regulate exocytotic secretion rather than autophagy[32], and the functional diversity among different Exo84 family members has been largely unappreciated.

Here, we demonstrate that a plant-unique Exo84 isoform, Exo84c, is implicated in autophagy through its interaction with VAP27 proteins. The *exo84c* knock-out mutant has a longer stigma life span and a corresponding higher seed set, which can be complemented with Exo84c:GFP, rather than Exo84a:GFP or Exo84b:GFP, suggesting that the three Exo84 isoforms exhibit some functional diversity. Increasing GFP:Exo84c and Sec6:GFP-labelled structures (autophagosome related) are co-accumulated with VAP27-labelled ER membranes in the vacuole in ageing stigmatic papilla cells, suggesting strong autophagy activity involved in senescent cells. Our work here provides insight into the molecular and cellular mechanism that regulates stigma senescence and flower receptivity and we show that this involves a VAP27-Exo84 mediated ER-autophagy pathway.

## Results

### Exo84c proteins interact with VAP27 and promote the formation of ER-membrane-derived punctae

The exocyst complex is essential for pollen-pistil interactions and for pollen acceptance[28,29]. In a recent proteomics screen, we identified several exocyst proteins that can interact with VAP27, and one of these proteins was Exo84c, a subunit from the exocyst complex[26,28–30,33]. The VAP27 protein is localized to the ER network and ER-PM contact sites, regulating multiple membrane trafficking events including endocytosis and autophagy[12,15,17,34–36]. It is well-known that the exocyst complex is involved in vesicle docking and secretion at the plasma membrane but its interaction with an ER membrane protein is surprising. Phylogenetic analyses indicated that the Exo84 family consists of three isoforms in plants (Fig. 1a) that can be divided into two groups. Interestingly, Exo84c is only found in higher plants, indicating that a further gene duplication event occurred that predates the diversion between mosses and higher plants with respect to Exo84a and b (Fig. 1a). Therefore, it is reasonable to suggest that functional diversity might exist among the different Exo84 isoforms.

The interaction between different Exo84 isoforms and VAP27-1/3 was tested using the bimolecular fluorescence complementation (BiFC) assay (Fig. 1b), and the expression of all fusion proteins was confirmed using immunoblots (Fig. 1d). Only cells expressing nYFP-Exo84c and cYFP-VAP27 showed strong fluorescent signals at punctate structures, suggesting the interaction between Exo84c and VAP27-1 or -3 are specific. These punctate structures were associated closely with ER and co-localized with the ER maker, CFP:HDEL (Fig. 1c,

Supplementary Fig. 1a), implying that the VAP27-Exo84c-labelled structures might derive from the ER. A similar phenomenon was found in the *Nicotiana benthamiana* transient expression system when Exo84c and VAP27-3 were co-expressed (Supplementary Fig. 1b–d); neither GFP:Exo84a or GFP:Exo84b co-localized with VAP27-3:mCh (as suggested by the Pearson coefficient analysis, Supplementary Fig. 1e), which is in agreement with the BiFC data (Fig. 1b). In addition, a co-IP assay was conducted using a stable Arabidopsis line expressing Exo84cp::Exo84c:GFP in *exo84c* mutant background. Using a VAP27 antibody that detects either VAP27-1 and -3, we confirmed that the endogenous VAP27 co-purified with Exo84c:GFP, indicating that the interaction between Exo84c and VAP27-3 also takes place endogenously (Fig. 1e). Here, the stable Arabidopsis line expressing Exo84b:GFP was used as the negative control, and it does not co-purify with VAP27 (Fig. 1e), further confirming that the interaction of Exo84c and VAP27 is unique among all Exo84 isoforms. Similar results were also confirmed using a one-on-one Y2H assay (Fig. 1f).

### The *exo84c* loss-of-function mutant shows delayed stigmatic senescence

To further investigate the biological function of Exo84c, the phenotype of the *exo84c* knock-out mutant was studied. Unlike the mutants of Exo84b or other exocyst subunits that have a pleiotropic effect on development, the *exo84c* mutant does not affect overall plant growth under normal conditions[37,38]. It has been reported that the acceptance of compatible pollen was reduced in the *exo84c* mutant at low humidity conditions, and this phenotype can be partially restored at high humidity conditions[39]. In our study, similar a phenotype was also observed with low humidity (Supplementary Fig. 2a), but no significant differences in pollen tube growth between Col-0 and *exo84c* were found under normal conditions (Fig. 2a). However, when pollination was performed using ageing flowers (72 HAE, hours after emasculation), an opposite phenotype was observed; more pollen germination on the *exo84c* mutant pistils compared to wild type was found (Fig. 2b, c). This phenotype was complemented by the Exo84cp::Exo84c:GFP construct transformed into the *exo84c* mutant background (*exo84c*/Exo84cp::Exo84c:GFP), suggesting that the phenotype is indeed related to the function of Exo84c, which then could be related to stigmatic tissue senescence.

To further investigate this, we used dual staining with FDA and PI to label live and dead papilla cells[5,40]. Live-cell imaging was conducted at 24, 48 and 72 HAE using the Col-0, *exo84c* mutant, GFP:Exo84c overexpression and *exo84c*/Exo84c:GFP Arabidopsis lines (Fig. 2d). As the dead cells were strongly labelled with PI but not with FDA, the ratio of live cells to dead cells at different time points could be calculated (Supplementary Fig. 2b). The results show that compared to Col-0 (79.28%), the numbers of live cells were significantly higher in the *exo84c* mutant (93.50%) at 72 HAE (Fig. 2e) and significantly lower in the GFP:Exo84c overexpressing plants (59.10%) (Fig. 2f). In *exo84c* complemented plants (*exo84c*/Exo84cp::Exo84c:GFP), the ratio of live to dead papilla cells is similar to that of the wild type (Fig. 2e). Thus, the higher pollen germination rate observed in the *exo84c* mutant (Fig. 2b) is likely related to the delayed senescence of stigmatic papilla cells.

In order to investigate whether the other Exo84 isoforms are also involved in stigma senescence, the Exo84c promoter was used to drive the expression of either Exo84a or Exo84b in the *exo84c* background. Exo84a:GFP and Exo84b:GFP localizes predominantly to the cytoplasm of papilla cells, which is a similar pattern to the localization of Exo84c:GFP (Supplementary Fig. 2c). Noteworthy is that plastids also produced strong auto-fluorescence in some papilla cells but this did not affect our analysis, as their structures were easily distinguishable (Supplementary Fig. 2c, arrow). Both Exo84a/b transgenic lines exhibited delayed stigmatic senescence (Fig. 2e, Supplementary Fig. 2d, e), similar to the phenotype of the *exo84c* mutant, suggesting that these constructs were not able to complement the phenotype of

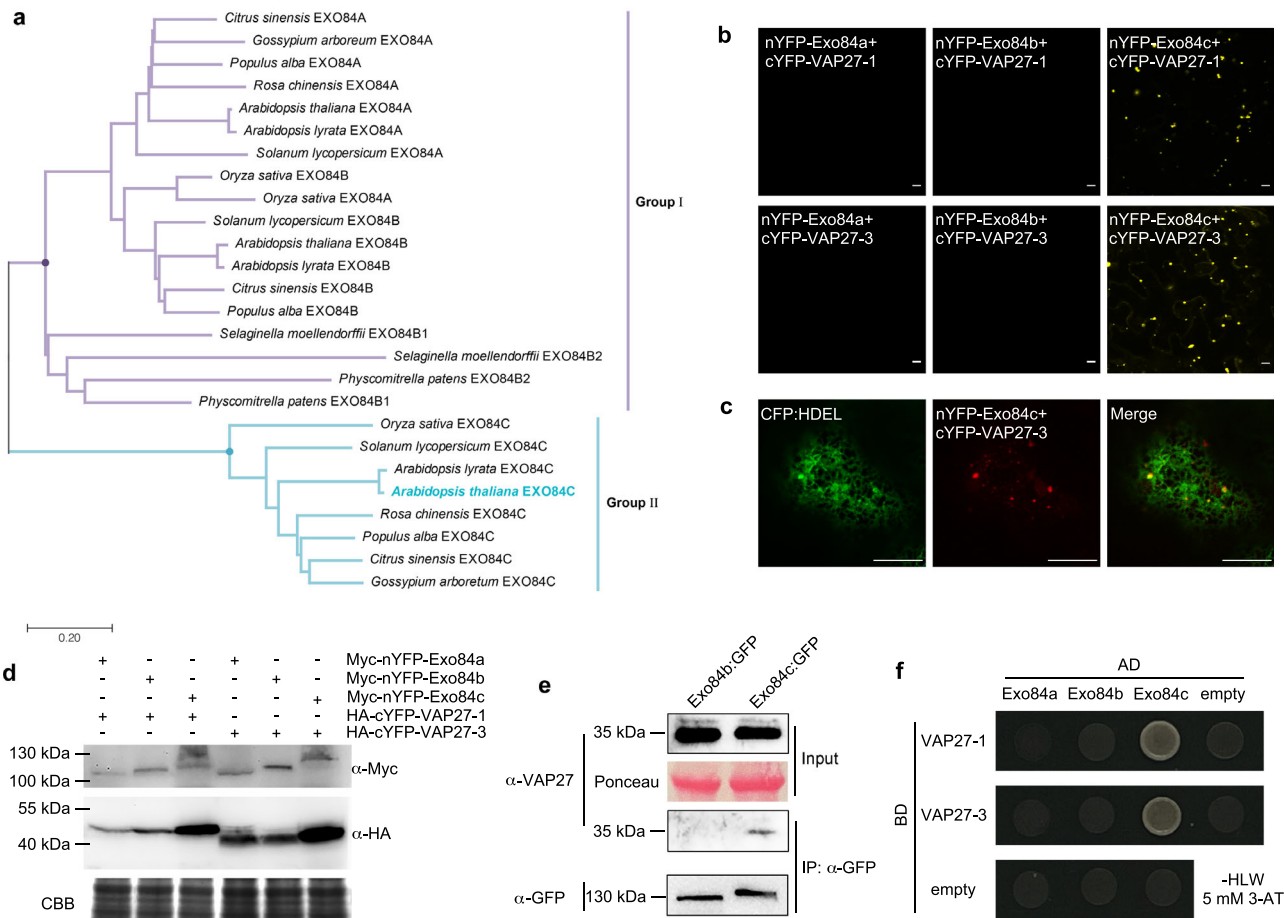

**Fig. 1 | VAP27 proteins interact specifically with Exo84c. a** Phylogenetic analysis of Exo84 isoforms in various plant species. The evolutionary history is inferred using the Neighbour-Joining method. **b** BiFC assay indicates an interaction between VAP27-1/-3 and Exo84c in *N. benthamiana*, but not with Exo84a and Exo84b. Scale bars, 10 μm. The experiment was repeated twice with similar results. **c** The Exo84c and VAP27-3 labelled punctate structures associated with CFP:HDEL-labelled ER network in *N. benthamiana*. Scale bars, 10 μm. The experiment was repeated twice with similar results. **d** Immunoblot analysis of the BiFC studies, showing all protein combinations were expressed successfully. Please note, the loaded protein for

*exo84c*. An immunoblot with a GFP antibody further confirmed that both Exo84a:GFP and Exo84b:GFP were expressed at similar levels as Exo84c:GFP (Supplementary Fig. 2f). Moreover, the *exo84a* mutant[38] was analyzed using the FDA-PI dual staining. The results show that knocking out Exo84a does not affect the stigma life span (Supplementary Fig. 3). Therefore, we suggest that Exo84c has a unique function in stigma senescence and that this function is different from that of the other Exo84 isoforms which are known to regulate secretion.

Furthermore, plants were manually pollinated with Col-0 pollen at 24, 48 and 72 HAE, respectively. Consistent with the delayed stigma degeneration results, the *exo84c* mutant set more seeds at 72 HAE compared to Col-0, while GFP:Exo84c overexpressing plants exhibit the opposite effect; no significant differences were found in the *exo84c* complemented plants (Fig. 2g, h). We suspect that the changes in stigmatic life span may partially affects the seed-set phenotype shown here.

### The *vap27-1/3* loss-of-function mutant shows early stigmatic senescence

As VAP27 interacts specifically with Exo84c, it is highly possible that VAP27 also participates in stigma senescence. To test this idea, two

Exo84c + VAP27 were two times higher than the other two samples in order to make the Myc-nYFP-Exo84c band visible (due to a higher degradation rate, discussed later). Coomassie brilliant blue (CBB) staining is used as the loading control. **e** Co-IP assay using *exo84c*/Exo84cp::Exo84c:GFP stable transgenic Arabidopsis confirms the interaction between VAP27-3 and Exo84c under endogenous conditions. Stable Arabidopsis plants transformed with Exo84bp::Exo84b:GFP was used as a negative control. The IP assay was repeated twice with similar results. **f** Y2H analysis shows that VAP27-1 and VAP27-3 interact specifically with Exo84c rather than Exo84a and Exo84b.

independent *vap27-1/3* CRISPR double knock-out lines were created (Supplementary Fig. 4a); both genes contain nucleotide deletions which result in frame shift or domain deletion mutations. Such knock-out effects were further confirmed by immunoblotting using a VAP27 antibody that detects VAP27-1 and -3 (Supplementary Fig. 4b–d). FDA/PI dual staining was conducted using Col-0 and *vap27-1/3* double mutants (Fig. 3a). The results show that compared to Col-0 (76.14%), the ratio of live to dead cells was significantly lower in both *vap27-1/3* double mutants (59.95% and 56.89%, respectively) at 72 HAE (Fig. 3b). The seed set of *vap27-1/3* double mutants was also quantified as before (Fig. 2h). Consistent with the stigma degeneration results, the *vap27-1/3* double mutants set fewer seeds at 72 HAE compared to Col-0 (Fig. 3c, d). These results were in stark contrast to the phenotype of the *exo84c* mutant, which showed delayed stigma senescence and a higher seed set.

Surprisingly, the phenotype of the *vap27-1/3* double mutant somehow mimics the effect of Exo84c overexpression, indicating that a regulatory machinery might exist that employs VAP27 and Exo84c. For example, when the function of VAP27 is partially perturbed, the protein level of Exo84c is elevated. Consequently, we analyzed the protein abundance and turnover rate of Exo84c in either VAP27 overexpressing or mutant plants.

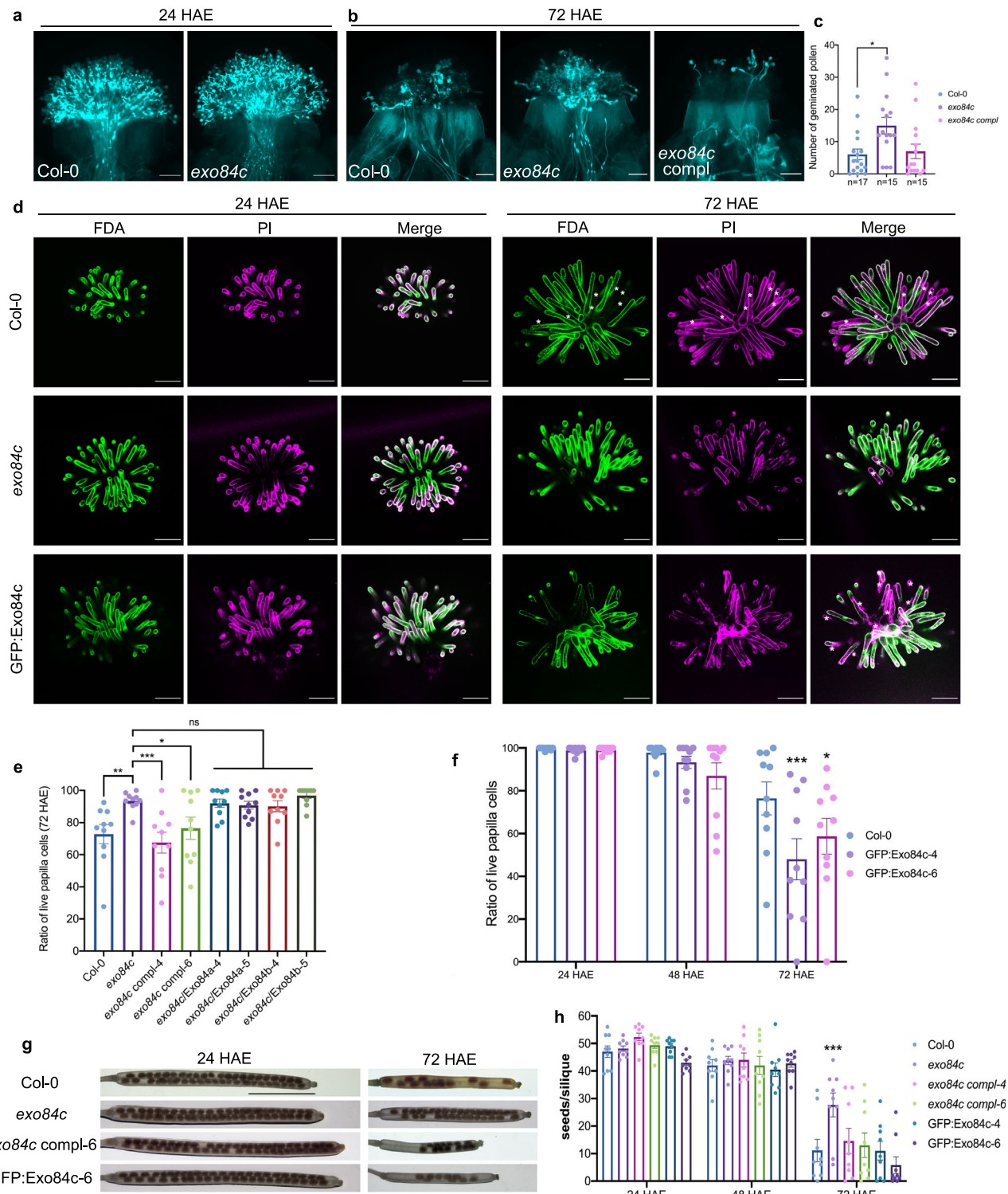

## VAP27 promotes Exo84c degradation, a process requiring the ER network

An Exo84c antibody was raised and it detected a single band of approximately 100 kDa in wild type Arabidopsis extracts; this band was absent from the *exo84c* mutant extracts (Supplementary Fig. 5), suggesting that the antibody is specific to Exo84c. In vivo protein degradation assays were then conducted using both *exo84c* and *vap27-1/3* mutants following cycloheximide (CHX) treatment, a protein biosynthesis inhibitor[41]. Compared to that in Col-0, the degradation of Exo84c was found to be much slower in *vap27-1/3* double mutants (Fig. 3e, f). Similarly, the degradation rate of VAP27-1/3 is also reduced

significantly in the *exo84c* mutant compared to that observed in Col-0 (Fig. 3g, h). Therefore, it is very likely that the endogenous level of Exo84c and VAP27 are precisely regulated by each other; altering the expression of one protein is likely to affect the protein abundance of the other. Furthermore, we used the transient *N. benthamiana* system to express Exo84c and VAP27-3, either alone or together. The average fluorescence intensity and protein level of either GFP:Exo84c or VAP27-3:mCh is significantly reduced when the two proteins are overexpressed at the same time (Supplementary Fig. 6a–d), in stark contrast to the result of their loss-of-function mutant. In cells co-expressing VAP27-3 with different Exo84 isoforms, the protein level of

**Fig. 2 | Exo84c regulates the life span of stigma. a, b** Pistils from different plants pollinated at 24 and 72 HAE are shown. **a** Aniline blue staining of pistils from Col-0 and *exo84c* plants pollinated with Col-0 pollen at 24 HAE. **b** Aniline blue staining of pistils from Col-0, *exo84c*, and *exo84c*/Exo84cp:Exo84c:GFP (*exo84c* complement) plants pollinated with Col-0 pollen at 72 HAE. Scale bars, 100 µm. **c** Quantification of pollen germination of Col-0, *exo84c*, and *exo84c*/Exo84cp:Exo84c:GFP plants following manual pollination with Col-0 pollen at 72 HAE. The efficiency of pollen germination is significantly higher in *exo84c* plants at 72 HAE compared to Col-0. n represents the number of stigmas used for quantification. Error bars represent SEM, and the asterisks represent means that are significantly different at $P = 0.0139$ from a one-way ANOVA with Tukey's multiple comparisons test. **d** Examples of live/dead papilla cells analysis using FDA/PI staining. Stigmatic papilla cell senescence indicated by FDA (green) and PI (magenta) staining at 24, 48 and 72 HAE. The asterisks represent dead cells. Scale bars, 100 µm. **e** Quantification of the live papilla cells from Col-0, *exo84c*, *exo84c* complemented plants (*exo84c*/Exo84cp::Exo84c:GFP, namely *exo84c* compl), *exo84c* complimented with Exo84cp::Exo84a:GFP and *exo84c* complimented with Exo84cp::Exo84b:GFP plants at 72 HAE. Ten stigmas

from three independent plants were used for the quantification ($n = 10$). Error bars represent SEM, and the asterisks represent means that are significantly different at $P < 0.05$ from a one-way ANOVA with Dunnett's multiple comparisons test. **f** Quantification of the live papilla cells from Col-0 and UBQ10::Exo84c:GFP over-expressing plants at 24, 48 and 72 HAE. The ratio of live papilla cells is much lower in UBQ10::GFP:Exo84c (overexpression) plants compared to Col-0. Ten stigmas from three independent plants were used for the quantification ($n = 10$). Error bars represent SEM, and the asterisks represent means that are significantly different at $P < 0.05$ from a two-way ANOVA with Dunnett's multiple comparisons test. **g** Representative images of cleared siliques used for quantification. Scale bar, 2 mm. **h** Quantification of seeds per silique from Col-0, *exo84c*, *exo84c*/Exo84cp::Exo84c:GFP, and UBQ10::GFP:Exo84c plants following manual pollination with Col-0 pollen at 24, 48 and 72 HAE. The seed sets are significantly higher in *exo84c* plants at 72 HAE compared to Col-0. Nine siliques from three independent plants were used in seed counting for each line ($n = 9$). Error bars represent SEM, and the asterisks represent means that are significantly different at $P < 0.05$ from a two-way ANOVA with Dunnett's multiple comparisons test.

GFP:Exo84c was always found to be lower than the other two isoforms in the presence of VAP27-3 (Fig. 3i). However, co-expression with VAP27-3 does not affect the average fluorescence intensity or protein level of Exo84b isoform when they are transiently expressed in *N. benthamiana* (Supplementary Fig. 6e–g), indicating that the degradation of Exo84 requires its interaction with VAP27, and this process is specific for Exo84c. As a result, the protein turnover of Exo84c may be reduced in the *vap27-1/3* double mutants, thereby generating the stigmatic and seed-set phenotypes that are similar to the Exo84c overexpressing line (Fig. 2d, f).

Interestingly, when Exo84c is co-expressed with a VAP27 truncation without the ER transmembrane domain, VAP27-3ΔTMD, the punctate structures were not formed (Supplementary Fig. 6h), and the fluorescence intensity and protein level of either GFP-Exo84c or VAP27-3ΔTMD:mCh were also not reduced (Supplementary Fig. 6i–k), suggesting that the ER targeting of VAP27 is essential for the degradation of Exo84c. Taken together, these results suggest that the interaction between Exo84c-VAP27 is likely to promote the degradation of themselves. The involvement of VAP27 and the ER membrane are essential for the formation of VAP27-Exo84c punctate structures and the concomitant selective degradation of both proteins, as indicated by the immunoblotting data (Supplementary Fig. 6l).

### Exo84c-VAP27-regulated degradation is elevated during stigma cell senescence

Whether the delayed stigma phenotype is related to the Exo84c and VAP27 interaction and degradation is the question that now arises. Stable Arabidopsis plants expressing both GFP:Exo84c and VAP27-3:mCh were generated by crossing, and the protein behaviours were analyzed at the subcellular level. In papilla cells either expressing GFP:Exo84c alone or co-expressing GFP:Exo84c and VAP27-3:mCh, their signals accumulated in the vacuole of papilla (Supplementary Fig. 7a) during stigma senescence. The numbers of vacuole-accumulated Exo84c puncta at 72 HAE increased significantly in cells co-expressing VAP27-3:mCh, compared with cells only expressing GFP:Exo84c (Fig. 4a–c). Such age-induced vacuole accumulation was also found when Exo84c is expressed at endogenous levels using the complemented line (Supplementary Fig. 7b). The intensity of vacuole-accumulated VAP27-3:mCh is also stronger in GFP:Exo84c and VAP27-3:mCh co-expressing cells, as indicated by the increased internal/cortical signal intensity ratio (Fig. 4d). These results suggested that Exo84c and VAP27 are likely to be transported to the vacuole for degradation in ageing stigma tissue.

Autophagy is known to regulate bulk degradation of proteins or organelles during senescence and plant reproduction[33,42–46]. Therefore, it is possible that the vacuole accumulation of GFP:Exo84c and VAP27-3:mCh is autophagy dependent. To test this idea, papilla cells co-

expressing GFP:Exo84c and VAP27-3:mCh were treated with concanamycin A (Conc A, a V-ATPase inhibitor that prevents vacuolar degradation), and the results show that GFP:Exo84c and VAP27-3:mCh co-localize at autophagosome-like structures in the vacuole (Fig. 4e). A similar co-localization was found in *exo84c*/Exo84cp::Exo84c:GFP + VAP27-3:mCh stable co-expression papilla cells following Conc A treatment (Supplementary Fig. 8a). Moreover, co-localization of GFP:ATG8a (an autophagosome marker) and VAP27-3:mCh was also found in the vacuoles following Conc A treatment, suggesting these VAP27-labelled punctate structures are indeed autophagosome related (Fig. 4f). Moreover, Arabidopsis plants expressing VAP27-3:mCh and GFP:Exo84c in the *atg5* mutant background were generated. No vacuole accumulation of GFP:Exo84c or VAP27-3:mCh was observed in these plants (Fig. 4g, Supplementary Fig. 7c), which further confirms that the vacuole accumulation of GFP:Exo84c and VAP27-3:mCh is through the autophagy pathway.

Interestingly, the accumulation of vacuole degradation products in papilla cells is seen even without Conc A treatment (Fig. 4a, b) and this effect is very different to that seen in cells from the vegetative tissue. This could be because the cytoplasm-to-vacuole transport in papilla cells is very active and as a result, some Exo84/VAP27 signal in the vacuole may be identified when the vacuole degradation machinery becomes saturated. Alternatively, the pH in papilla cells may increase gradually after 24 HAE (as a result of senescence), so the GFP signal accumulated and became visible.

Since the stigmatic tissue is fragile, it is hard to perform drug treatment and immunoblot analysis. We, therefore, studied the localization of Exo84c and VAP27 in root cells as an alternative approach to classical autophagic assays. In Arabidopsis co-expressing GFP:ATG8a and VAP27-3:mCh, GFP:ATG8a-labelled autophagosomes co-localize with VAP27-3:mCh in the vacuole after Conc A treatment (Supplementary Fig. 8b). The co-localization between VAP27 and autophagosomes was further confirmed using the Pearson coefficient analysis (Supplementary Fig. 8c). Meanwhile, in an autophagy mutant (*atg5*), no Exo84c or VAP27-3 signal was observed in the vacuole after Conc A treatment (Supplementary Fig. 8d–g). Moreover, the vacuole-accumulated GFP:Exo84c signal in the presence of VAP27-3:mCh was found to be significantly higher when compared with the cells only expressing GFP:Exo84c upon Conc A treatment (similar to their behaviours in papilla cells, Fig. 4b), suggesting that the autophagic degradation of GFP:Exo84c is elevated in the presence of VAP27-3:mCh (Fig. 4h, i). This result was further confirmed by immunoblotting, where the protein level of Exo84c is lower in the presence of VAP27-3, but this effect is restored after Conc A treatment (Fig. 4j). Taken together, we speculate that the phenomenon we observed in senescent papilla cells (Fig. 4a, b) is likely caused by VAP27-3 regulated Exo84c turnover.

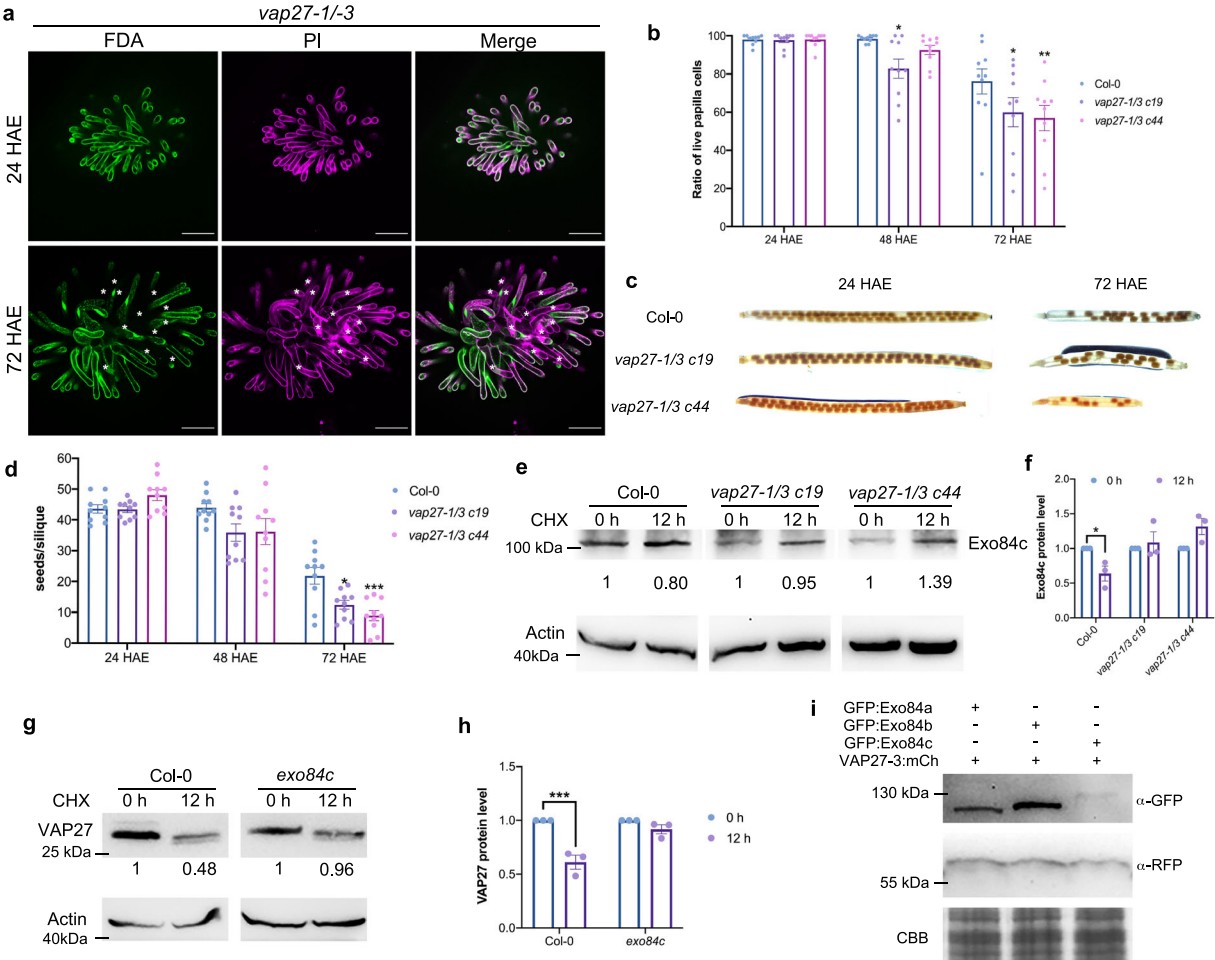

**Fig. 3 | The *vap27-1/3* mutant showed an early stigma senescence phenotype.**
**a** Representative images of live/dead papilla cells in Col-0 and *vap27-1/3* mutant plant at 72 HAE. Scale bars, 100 μm. **b** Quantification of the live papilla cells from the *vap27-1/3* mutant, the ratio of live to dead papilla cells is significantly lower than in the wild type plants (*n* = 10 stigmas). Error bars represent SEM, and the asterisks represent means that are significantly different at *P* < 0.05 from a two-way ANOVA with Dunnett's multiple comparisons test. **c, d** Quantification of seeds per silique from Col-0 and two *vap27-1/3* mutant plants following manual pollination with Col-0 pollen at 24, 48 and 72 HAE. The seed sets are significantly lower in the *vap27 mutant* plants at 72 HAE compared to Col-0 (*n* = 10 siliques). Error bars represent SEM, and the asterisks represent means that are significantly different at *P* < 0.05 from a two-way ANOVA with Dunnett's multiple comparisons test. **e** The degradation of endogenous Exo84c in *vap27-1/3* Arabidopsis CRISPR mutants is inhibited as detected by immunoblotting analysis. Two independent *vap27-1/3* CRISPR mutants were detected with similar results. Band intensities were quantified relative to the protein amount of Col-0 and *vap27-1/3* mutants at 0 h, respectively. Actin was used as a loading control. **f** Quantification of Exo84c protein levels in *vap27-1/3* CRISPR mutants. Three biological repeats were taken for the quantification. Error

bars represent SEM, and the asterisks represent means that are significantly different at *P* < 0.05 from a two-way ANOVA with Sidak's multiple comparisons test. **g** Immunoblotting analysis indicating that the endogenous VAP27 degradation is inhibited in the Arabidopsis *exo84c* mutant. Three independent experiments were performed with similar results. In both **e** and **g**, samples were pre-treated with CHX to block protein translation. Band intensities were quantified relative to the protein amount of Col-0 and *exo84c* at 0 h and are indicated underneath the blot, respectively. Actin levels were detected as loading controls with an anti-actin antibody. **h** Quantification of VAP27 protein levels in the *exo84c* mutant. Three biological repeats were taken for the quantification. Error bars represent SEM, and the asterisks represent means that are significantly different at *P* < 0.05 from a two-way ANOVA with Sidak's multiple comparisons test. **i** Immunoblot analysis of *N. benthamiana* leaf cells transiently expressing different Exo84 isoforms with VAP27-3. Coomassie brilliant blue (CBB) staining is used as the loading controls. The experiment was repeated twice with similar results. Please note, the immunoblots of **e** and **g** are from the same membrane it was cropped to fit the figure panel, original images can be found in Source data.

## Exo84c-VAP27 positive punctate structures are autophagosome related

Since the VAP27-mediated Exo84c degradation is likely to be autophagy dependent, we then asked whether the Exo84c-VAP27-labelled punctate structures were autophagosome related. In the stable Arabidopsis line, GFP:Exo84c localized to the cytoplasm and to punctate structures in root elongation cells, and VAP27-3:mCh localized to the ER network (Fig. 5a). Interestingly, Exo84c exhibited strong co-localization with VAP27-3 when both proteins were co-expressed (Fig. 5b, Supplementary Fig. 9a), which is in agreement with this phenomenon in *N. benthamiana* (Supplementary Fig. 1c). Similar structures were also found in ageing papilla and style cells co-expressing

GFP:Exo84c and VAP27-3:mCh (Fig. 5c, d), indicating that those punctate structures are induced by the senescence of stigma tissue. These structures exhibit strong association with the VAP27-labelled ER network and VAP27 positive punctate structures, whereas no co-localization between Exo84c and VAP27-3ΔTMD was found (Supplementary Fig. 9b–d); kymograph analysis confirmed that GFP:Exo84c and VAP27-3:mCh co-localize at ER-membrane-derived puncta, which are either static or mobile during the time of imaging (Supplementary Fig. 9e).

We co-expressed the BiFC constructs, nYFP-Exo84c and cYFP-VAP27 with different organelle markers, and found that these VAP27-Exo84c-labelled puncta strongly co-localize with the ER marker,

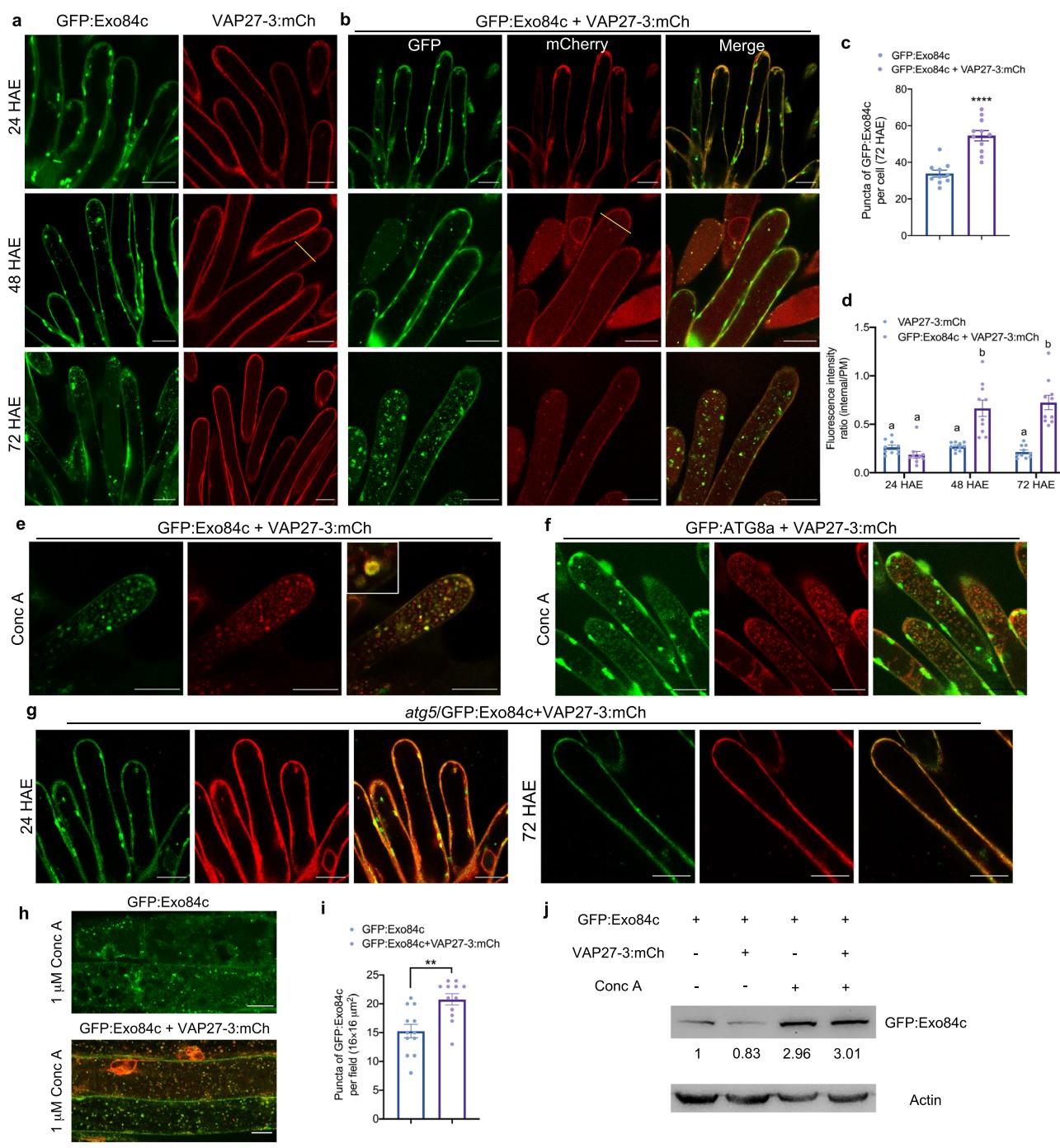

CFP:HDEL, and the autophagosome marker, CFP:ATG8e (Fig. 5e, f). Therefore, we suspected that the interaction between Exo84c and VAP27 may take place at the ER and autophagosome interface, which is maintained by VAP27 in plants[22]. Thus, these Exo84c-VAP27 positive structures might be ER-autophagosome related, and their interaction may take place during autophagosome biogenesis. When CFP:ATG8e was co-expressed with mCh:Exo84c and VAP27-3:mCh in *N. benthamiana*, the protein level of CFP:ATG8e (Fig. 5g, h) is reduced, accompanied by an accumulation of free CFP. In contrast CFP:ATG8e does not degrade when it is co-expressed with RFP:Exo84b or VAP27-3:mCh (Supplementary Fig. 9f), indicating that a stronger autophagic degradation activity is induced in the presence of both Exo84c and VAP27.

In stable transgenic Arabidopsis root cells, Conc A treatment caused numerous GFP:Exo84c and VAP27-3:mCh co-labelled punctae

to be localized to the vacuole (Fig. 5i, j). These structures were ER-derived and autophagosome related, as indicated by high magnification light-microscopy using an autophagosome marker CFP:ATG8e (Fig. 5m) and TEM studies (Fig. 5l), reminiscent of the ER-autophagosome structures previously observed in Arabidopsis and *N. benthamiana* (Fig. 5b, f). In the presence of both Conc A and wortmannin (Wort), an inhibitor of plant PI3Ks and the autophagy pathway, no vacuolar accumulation of GFP:Exo84c or VAP27-3:mCh puncta was found, indicating that both Exo84c and VAP27-3 are likely degraded in the autophagy process (Fig. 5k).

## Exo84c is essential for the degradation of exocyst complex-labelled structures

In plants and yeasts, most exocyst subunits form a high-molecular weight complex that can be co-purified[23,47]. We therefore, suspected

**Fig. 4 | Exo84c positive structures accumulate in vacuoles through the autophagy pathway during stigmatic papillae senescence. a, b** The vacuole accumulation of Exo84c and VAP27-3 in papilla cells at 24, 48 and 72 HAE from stable transgenic Arabidopsis lines expressing UBQ10::GFP:Exo84c **a**, VAP27-3p::VAP27-3:mCh **a** or UBQ10::GFP:Exo84c + VAP27-3p::VAP27-3:mCh **b** GFP:Exo84c-labelled punctate structures were detected in the vacuole from 48 HAE. GFP:Exo84c significantly accumulates in the vacuole only when it is co-expressed with VAP27-3:mCh. Scale bars, 20 μm. **c** Quantification of the number of GFP:Exo84c-labelled punctae in papilla cells at 72 HAE. GFP:Exo84c punctae number significantly increases in plants co-expressing UBQ10::GFP:Exo84c and VAP27-3p::VAP27-3:mCh (n = 10 papilla cells from three individual stigmas). Error bars indicate the SEM, and the asterisk represents means that are significantly different at P < 0.05 from a two-tailed Student's *t*-test. **d** Quantification of the fluorescence intensity ratio of internal/cortical in papilla cells. The vacuole accumulation of VAP27-3:mCh significantly increases at 48 and 72 HAE in Arabidopsis expressing UBQ10::GFP:Exo84c + VAP27-3p::VAP27-3:mCh (n = 10 papilla cells from three individual stigmas). Error bars represent SEM, and letters above the bars indicate means that are significantly different at *P* < 0.05 from a two-way ANOVA with Tukey's multiple comparisons test. **e** Papilla cells expressing GFP:Exo84c and VAP27-3:mCh were treated with Conc A for 8 hours, and strong accumulation of Exo84c and VAP27-3 co-labelled punctate structures were found in the vacuole. Scale bars, 20 μm. The treatment was repeated two times with similar results. **f** Similar effect was found in Conc A treated papilla cells expressing GFP:ATG8a and VAP27-3:mCh. Scale bars, 20 μm. The treatment was repeated two times with similar results. **g** The vacuole accumulation of Exo84c and VAP27-3 is blocked in the *atg5* mutant. Scale bars, 20 μm. The experiment was repeated two times with similar results. **h** Root cells of 5-day-old transgenic plants (UBQ10::GFP:Exo84c or UBQ10::GFP:Exo84c + VAP27-3p::VAP27-3:mCh) subjected to Conc A treatment. Scale bars, 10 μm. **i** Quantification of the number of vacuolar GFP:Exo84c dots in root cells following 1 μM Conc A treatment. GFP:Exo84c puncta number significantly increases when UBQ10::GFP:Exo84c is stably co-expressed with VAP27-3p::VAP27-3:mCh. Error bars indicate the SEM, twelve cells from three independent plants (n = 12) are used for the quantification; the asterisk represents means that are significantly different at *P* < 0.05 from a two-tailed Student's *t*-test. **j** Immunoblots showing GFP:Exo84c levels using a GFP antibody. The protein level is reduced in the presence of VAP27-3:mCh of stable transgenic Arabidopsis, indicating a higher turnover rate. Band intensities indicated underneath the blot were quantified relative to the band of GFP:Exo84c single expressing line without Conc A treatment, two independent experiments were performed with similar results.

that the exocyst complex may stay associated with Exo84c throughout the process of autophagy. In *N. benthamiana*, both Exo70B1:mCh and Sec10:mCh (two representative members of the exocyst complex) are recruited to the ER-autophagosome structures that are labelled by Exo84c and VAP27 (Supplementary Fig. 10a, c), and accumulated in the vacuole together with Exo84c-VAP27-labelled punctae upon Conc A treatment (Supplementary Fig. 10b). However, the Exo84a isoform was not able to recruit other exocyst subunits to such autophagosome structures in the presence of VAP27 (Supplementary Fig. 11). Therefore, we suspect that the expression of Exo84c and VAP27 not only affects their degradation but also likely regulates the turnover of other exocyst subunits. To test this hypothesis, we used Sec6, a core component of the exocyst complex that is commonly used as a marker for the exocyst complex[48,49]; stable Arabidopsis lines expressing Sec6:GFP were crossed into VAP27-3:mCh plants and the *exo84c* mutant backgrounds for further analysis.

In stigma tissues, Sec6:GFP-labelled puncta and VAP27-3:mCh were found in close contact (Supplementary Fig. 12). Conc A treatment was employed to investigate Sec6:GFP behaviour in both stigma tissue and root cells, and we found that the vacuole-accumulated Sec6:GFP signal was associated with VAP27-3:mCh (Fig. 6a–c), indicating that the degradation of the Sec6-positive compartment requires the ER to be the autophagosome membrane donor. To further investigate the function of Exo84c in exocyst degradation in papilla cells, the vacuole accumulation of Sec6-positive compartments was analyzed. The number of Sec6:GFP puncta accumulated in the vacuole in unpollinated papilla cells was found to be significantly reduced in the *exo84c* mutant compared to that in Col-0 plants (Fig. 6d–f), indicating that the degradation of exocyst compartments (likely be autophagosome related) or the turnover of the exocytotic complex requires Exo84c. Interestingly, a significant reduction of vacuole-accumulated Sec6:GFP signal was also found in the *exo84c* mutant compared to the wild type in root cells (Fig. 6g, h). These results are confirmed using immunoblotting which shows that the degradation of Sec6 is also reduced in the *exo84c* mutant (Fig. 6i), suggesting that the function of Exo84c (and likely in combination with VAP27) is in regulating the turnover of other exocyst subunits and exocyst-labelled structures (possibly autophagosomes).

Taken together, our results reveal an autophagy pathway regulated by Exo84c and VAP27 working in concert with the ER network and the exocytotic machinery (Fig. 7a). The vacuole accumulation of Exo84c and Sec6-labelled exocyst complexes/compartments is induced during senescence in stigma tissue, and the activity of Exo84c is required for this process (Fig. 7b). The *exo84c* loss-of-function mutant exhibits delayed stigma senescence and a prolonged effective

pollination period, suggesting that Exo84c-regulated autophagy is essential for stigma senescence during plant reproduction.

## Discussion

The functional diversity of the plant exocyst complex has mainly been focused on the Exo70 family to date, and the contribution from the Exo84 family is not clear. It has been suggested that plant Exo84s do not exhibit an autophagic function as its homologues in animal cells do, and that an alternative regulatory pathway (regulated by Sec5a and Rop8) is found in plants[32]. However, we show a plant-specific regulation of exocytotic material degradation through the interaction between Exo84c and VAP27s and suggest that the functional diversity amongst the Exo84 isoforms could provide another level of regulation which is in addition to the known functional diversity of the Exo70s. We suggest that Exo84c performs a unique function through its specific interaction with VAP27 during stigma degeneration. To further sustain this claim, we have performed genetic complementation studies of the *exo84c* mutant using Exo84b:GFP and Exo84a:GFP. The results demonstrate that neither can complement the delayed stigma senescence phenotype of the *exo84c* mutant (Fig. 2e).

The input from the ER network is also important for Exo84c-regulated autophagy, through interacting with VAP27, a key player in mediating the ER-PM interaction. Therefore, it is reasonable to speculate that the interaction between Exo84c and VAP27 facilitates a transient interaction between a PM-tethered exocyst complex and the ER membrane, driving ER-phagy biogenesis that originates from the ER[11,12]. Indeed, we found that the majority of the Exo84c-labelled structures are in contact with the cortical ER network and overlapped with VAP27-3 labelled puncta at the cell periphery, whereas no co-localization between Exo84c and VAP27-3ΔTMD was found (Supplementary Fig. 8a–e).

VAP27 and Exo84c-labelled compartments co-localize with the autophagosome marker, ATG8 and several exocyst subunits, were also recruited to these structures. Together with some previously characterized exocyst-mediated autophagy pathways, Exo84c-VAP27 are predicted to work in concert to regulate the selective degradation of exocyst compartment structures. For example, it has been reported that several Exo70 isoforms directly interact with ATG8 and mediate the selective autophagic degradation of secretory vesicles[30] or type-A ARR protein[31]. These findings raise the possibility that these Exo70 isoforms may interact with Exo84c-VAP27 on the one hand and, also recruit ATG8 to regulate the formation of intact autophagosomes. In a recent study, VAP27 is found as part of the autophagy machinery that links the ER membrane and autophagosomes[40]. We suspected that VAP27 may also act as an adaptor here that recruits the exocytotic

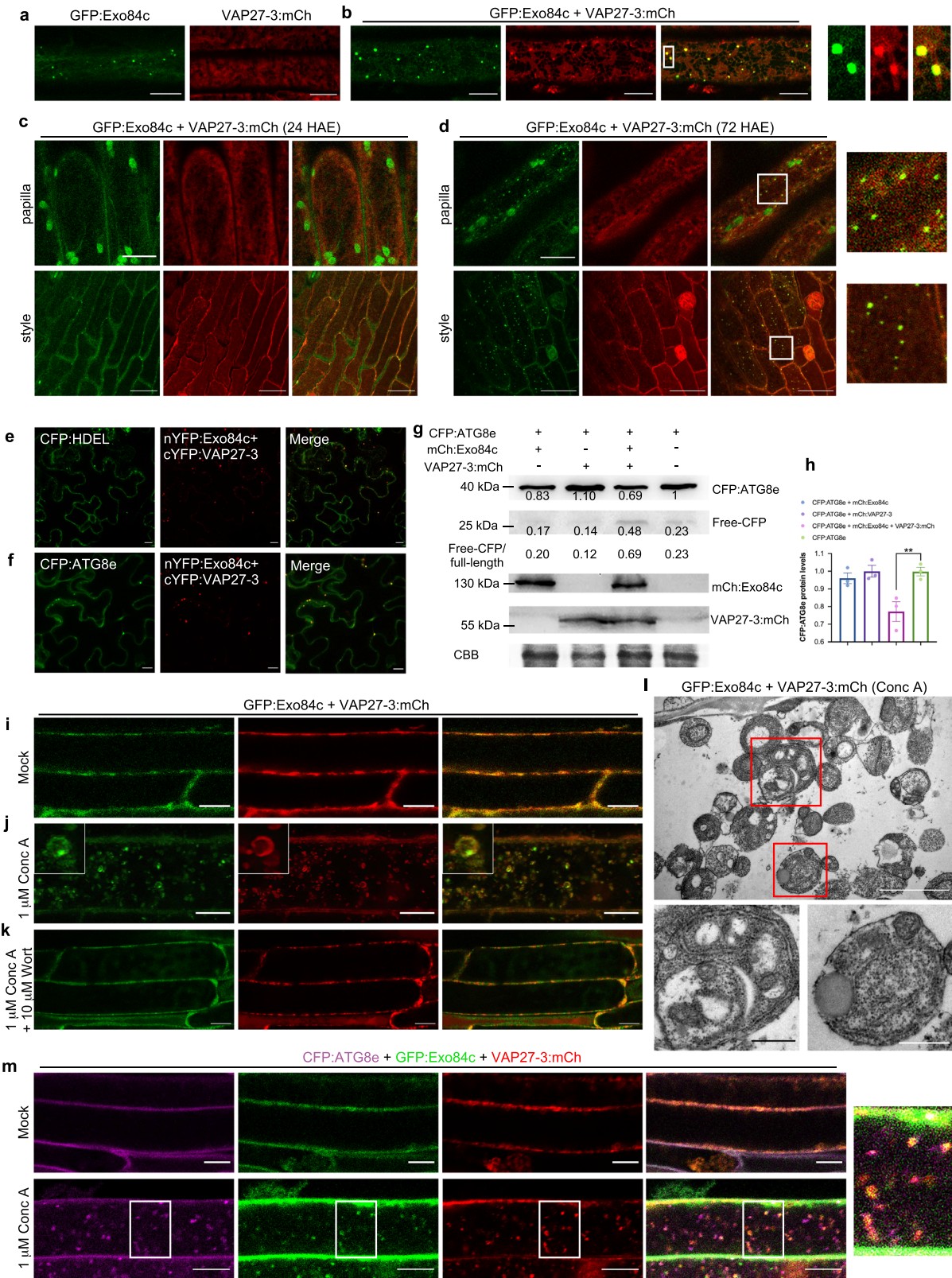

compartment and the core autophagy complex, facilitating the degradation of the exocytotic compartment. During this process, the ER/EPCS localized VAP27 may initiate the biogenesis of ER-derived autophagosomes through interacting with other autophagy receptors/regulators, such as NBR1, ULK1/ATG1 and WIPI2/ATG18[18,32,50], that are known to be associated with the exocyst complex. Removing Exo84c from the complex may affect the turnover of the exocyst complex and

indirectly affect the recycling of secretory cargoes. Alternatively, knocking out Exo84c may inhibit selective autophagy that regulates stigmatic tissue senescence.

It is known that ER-autophagosomes can be caused by ER stress in eukaryotes[51–53]. Studies have shown that the redox homeostasis in ER is disturbed during cell senescence in mammalian cells, usually inducing ER stress, which triggers the unfolded protein response (UPR) and

**Fig. 5 | The interaction between Exo84c and VAP27-3 regulates their degradation through the autophagic pathway. a**, **b** Localization of UBQ10::GFP:Exo84c and VAP27-3p::VAP27-3:mCh in root elongation region from stable Arabidopsis expressing UBQ10::GFP:Exo84c (**a**); VAP27-3p::VAP27-3:mCh (**a**); UBQ10::GFP:Exo84c and VAP27-3p::VAP27-3:mCh (**b**). Please note GFP:Exo84c-labelled punctate structures co-localized with the VAP27:mCh labelled ER puncta at the cortical section (zoom in). Scale bars, 10 μm. The experiment was repeated two times with similar results. **c**, **d** Localization of UBQ10::GFP:Exo84c and VAP27-3p::VAP27-3:mCh in papilla cells (upper panels) and style cells (lower panels) at 24 (**c**) and 72 HAE (**d**), respectively. Scale bars, 10 μm. The experiment was repeated two times with similar results. **e** The ER marker CFP:HDEL localizes to the punctate structures that are also labelled with Exo84c and VAP27-3 (similar to their localization in Arabidopsis). Scale bars, 10 μm. **f** VAP27-3 and Exo84c-labelled punctae structures recruit CFP:ATG8e, an autophagosome marker. Scale bars, 10 μm. The experiments in **e** and **f** were repeated two times with similar results. **g** The level of CFP:ATG8e is significantly reduced when it is co-expressed with VAP27-3:mCh and mCh:Exo84c in *N. benthamiana* cells, as indicated by the accumulation in free-CFP levels and the ratio of free-CFP/full-length protein. CBB staining is used as the loading control. The protein levels are normalized to the amount of CFP:ATG8e in the single expression sample (**e**–**g** are studied using the transient *N. benthamiana* system). **h** Quantification of CFP:ATG8e protein levels in *N. benthamiana* cells expressing VAP27-3:mCh alone, mCh:Exo84c alone, VAP27-3:mCh and mCh:Exo84c. Band intensities were quantified relative to the protein amount. Three biological repeats were taken for the quantification. Error bars indicate the SEM, the asterisks represent means that are significantly different at $P < 0.05$ from a one-way ANOVA with Dunnett's multiple comparisons test. Root cells of 5-day-old transgenic Arabidopsis seedlings expressing UBQ10::GFP:Exo84c and VAP27-3p::VAP27-3:mCh upon mock treatment **i**, 1 μM Conc A treatment **j**, or 10 μM wortmannin + 1 μM Conc A treatment **k**. Scale bars, 10 μm. The experiments were repeated two times with similar results. **l** Ultrastructural studies of Conc A treated root cells stably expressing UBQ10::GFP:Exo84c and VAP27-3p::VAP27-3:mCh (as in **j**). ER-derived structures are very abundant inside the vacuole. Scale bar, 2 μm (upper panel) or 500 nm (lower panel). Two biological repeats were taken with similar results. **m** Root cells of 5-day-old transgenic Arabidopsis seedlings expressing 35S::CFP:ATG8e, UBQ10::GFP:Exo84c and VAP27-3p::VAP27-3:mCh upon mock treatment (upper panels) and 1 μM Conc A treatment (lower panels). Scale bars, 10 μm. The experiment was repeated two times with similar results.

protective machinery to restore ER homeostasis[54,55]. However, persistent ER stress will impact cell fate, by inducing autophagy, apoptosis or senescence[54–56]. We suggest that a similar ER stress response also exists in ageing papilla cells. In this ER stress response, Exo84c promotes the formation of ER-derived autophagosomes by interacting with VAP27, which possibly enhances stigma senescence. Another possible explanation is, the senescence of stigma cells may require secretion of cell wall digesting enzymes, the knock-out of Exo84c may affect this process, therefore prevent the process of PCD.

In conclusion, we have uncovered a mechanism that works to regulate cell senescence and stigma life span. From the plant production perspective, this Exo84c-VAP27-mediated autophagy mechanism in floral stigma senescence could potentially have evolved to regulate the effective pollination period (EPP). Knocking out Exo84c or specifically blocking this autophagy process may prolong the stigma life span, with the delayed senescence of the stigma directly influencing the EPP thereby ensuring successful fertilization and ultimately determining the final yield of seeds or fruits[57].

## Methods

### Plant materials and growth conditions
Arabidopsis plants were grown in a greenhouse at 22 °C with 16 h of light and 8 h of darkness. Seeds of T-DNA insertion lines for Exo84a (*exo84a*: SALK_072277), Exo84c (*exo84c*: SALK_011569) and ATG5 (*atg5*: SAIL_129_B07) were obtained from the Nottingham Arabidopsis Stock Center. The stable transgenic VAP27-1p::VAP27-1:GFP, VAP27-3p::VAP27-3:GFP and UBQ10::GFP:ATG8a plants used in this study were previously published[58,59]. *N. benthamiana* plants were grown with 16 h of light (25 °C) and 8 h of darkness (18 °C). The *vap27-1/3 c19* and *vap27-1/3 c44* double knock-out mutants were generated using the CRISPR system. The synthetic guide RNA sequences from exons of VAP27-1 and VAP27-3 genes were designed with the web tool CRISPR-P2.0 (http://crispr.hzau.edu.cn/CRISPR2/)[60]. One single guide RNA (sgRNA) was chosen to target each gene. The sgRNAs cassette was finally cloned into the pHEE401E vector[61].

### Plasmid construction and plant transformation
The entry clones and fluorescent protein constructs for VAP27-1, -3 and VAP27-3ΔTMD were described in a previous publication[58]. The full-length coding sequence of Exo84c was cloned into pMDC43 and pUBN by Gateway reaction (Thermo Fisher Scientific) to generate 35S::mCherry:Exo84c and UBQ10::GFP:EXO84C, respectively. The Exo84c promoter (2-kb upstream of the start codon) and its genomic sequence, were cloned into pMDC107 to generate Exo84cp::Exo84c:GFP by Gateway reaction. The Exo84c promoter was also fused with the CDS of Exo84a or Exo84b, and cloned into pMDC107 to generate Exo84cp::Exo84a:GFP and Exo84cp::Exo84b:GFP by Gateway reaction, respectively. The 1.5-kb Sec6 promoter fused to its full-length Sec6 CDS (without stop codon) was cloned into pUBC to generate Sec6p::Sec6:GFP. Coding sequences of Exo84a, Exo84b, Sec10 and Exo70B1 were all cloned into either pMDC43 or pMDC83 vectors driven by 35S promoter. The detailed primers and construct information can be found in the Supplementary Tables 1 and 2.

All expression constructs were transformed into *Agrobacterium tumefaciens* GV3101 by electroporation. Transient transformation in *N. benthamiana* was conducted using leaf infiltration mediated by *A. tumefaciens* as reported before[62]. Five-week-old *N. benthamiana* plants were used for infiltration. Agrobacterium carrying different constructs was cultured in Luria-Bertani liquid medium at 28 °C overnight. And the agrobacteria were resuspended in infiltration buffer containing 100 μM acetosyringone, 50 mM MES and 2 mM Na₃PO₄. Stable transgenic Arabidopsis lines were generated by floral dipping[63].

### Phylogenetic analysis
The amino acid sequences of 26 Exo84 isoforms from diverse species were aligned to identify the evolutionary relationships. The evolutionary history was inferred using the Neighbour-Joining method[64]. The optimal tree with the sum of branch length = 8.07762086 is shown. The percentage of replicate trees in which the associated taxa clustered together in the bootstrap test (1000 replicates) are shown next to the branches[65]. The tree is drawn to scale, with branch lengths in the same units as those of the evolutionary distances used to infer the phylogenetic tree. The evolutionary distances were computed using the JTT matrix-based method[66] and are in the units of the number of amino acid substitutions per site. This analysis involved 28 amino acid sequences. All positions with less than 50% site coverage were eliminated, i.e., fewer than 50% alignment gaps, missing data, and ambiguous bases were allowed at any position (partial deletion option). There was a total of 766 positions in the final dataset. Evolutionary analyses were conducted in MEGA X[67].

### Production of polyclonal antibodies
The VAP27 antibody used in this study was generated according to the previous publication[21,33]. A VAP27-1 peptide (corresponding to amino acids 47–212 of *A. thaliana* Exo84c) was cloned into pET28a plasmid (Novagen, Darmstadt, Germany) that carries an N-terminal His-tag. An Exo84c specific (corresponding to amino acids 1-130 of *A. thaliana* Exo84c) peptide was cloned into pET28a plasmid (Novagen, Darmstadt, Germany) that carries an N-terminal His-tag. The expression of recombinant proteins was performed using the *Escherichia coli* BL21 strain and purified with nickel agarose beads

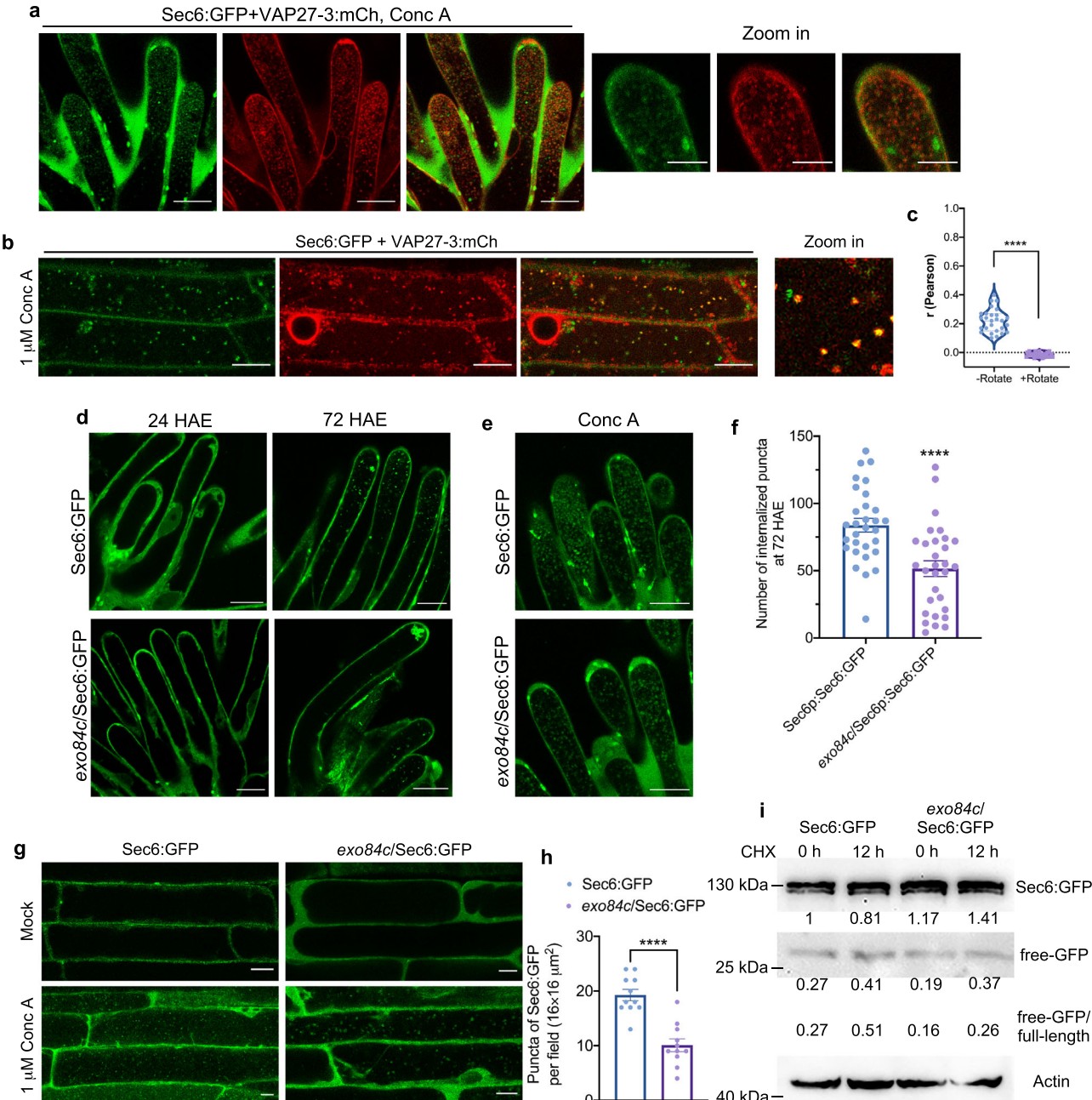

**Fig. 6 | Sec6-labelled exocyst complex or membranes accumulate in vacuole through Exo84c. a** Upon Conc A treatment, the number of vacuole accumulated Sec6 and VAP27-3 labelled structures is very abundant in ageing papillae cells. Scale bars, 20 μm. **b** Root cells from 5-day-old Arabidopsis lines expressing Sec6p::Sec6:GFP and VAP27-3p::VAP27-3:mCh subjected to 1 μM Conc A treatment. Scale bars, 10 μm. **c** Pearson coefficient analysis of the co-localization between Sec6 and VAP27 inside the vacuole (as in **d**, *n* = 27 cells). The Pearson's *r* values of merged images were quantified. For the negative control, the mCherry channel was rotated for 90°, the significant difference was determined by Student's *t*-test, *P* < 0.05. **d** The vacuole accumulation of Sec6p::Sec6:GFP in papillae cells at 24 and 72 HAE in Col-0 and *exo84c* genetic background, respectively. Scale bars, 20 μm. **e** The accumulation of Sec6:GFP in papilla cells after Conc A treatment. **f** The number of vacuole-accumulated Sec6:GFP punctae significantly decreases at 72 HAE in the *exo84c* genetic background. Thirty papilla cells from three stigmas were used for

the quantification (*n* = 30). Error bars represent SEM, and the asterisks represent means that are significantly different at *P* < 0.05 from a two-tailed Student's *t*-test. **g** Root cells from Arabidopsis expressing Sec6p::Sec6:GFP, Sec6p::Sec6:GFP in *exo84c* background upon mock treatment (upper panel), or 1 μM Conc A treatment (lower panel). Scale bars, 10 μm. **h** The number of vacuole-accumulated GFP-positive spots in *exo84c* root cells expressing Sec6p::Sec6:GFP significantly reduces than that in Col-0 root cells after 1 μM Conc A treatment. Error bars indicate the SEM, eleven cells from three independent plants (*n* = 11) are used for the quantification, the asterisks represent means that are significantly different at *P* < 0.05 from a two-tailed Student's *t*-test. **i** A western blot assay indicating that the turnover of Sec6:GFP is partially inhibited in the *exo84c* mutant of Arabidopsis after CHX treatment. Band intensities were quantified relative to the protein amount of Col-0 at 0 h and are indicated underneath the blot. Actin levels were detected as a loading control.

(BBI Life Sciences, China). Antibodies were raised in mice as previously described[68], purified proteins were dialyzed in PBS + 10% glycerol overnight at 4 °C. Polyclonal antibodies were raised in 4-week-old mice. Fifty micrograms antigen was used for each boost,

and a total of 4 boosts over 2 months were performed. Antiserum was collected 10 day after the final boost. Approval was granted by Animal Experimental Ethical Inspection of Laboratory Centre at Huazhong Agricultural University.

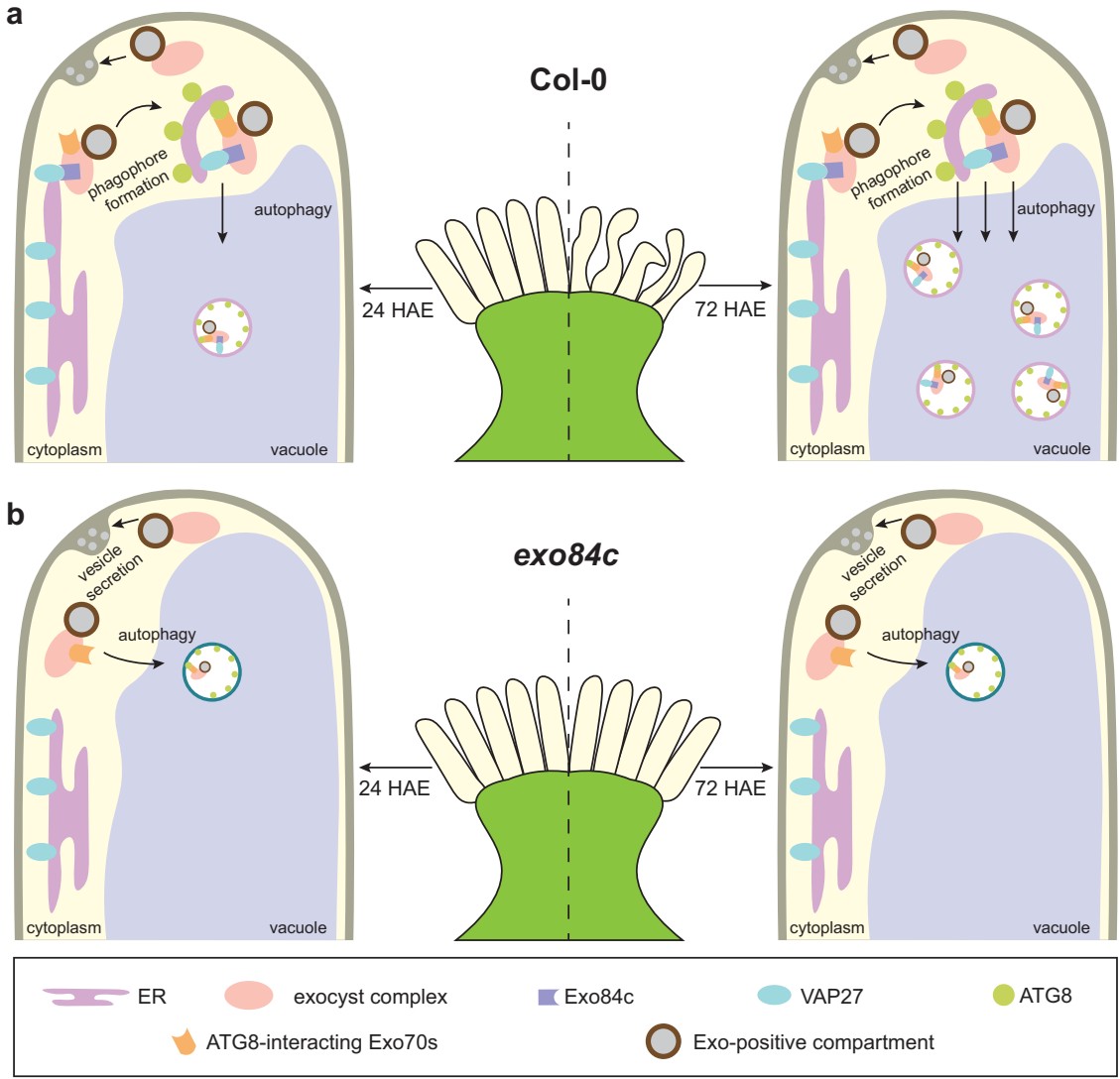

**Fig. 7 | A working model showing the possible mechanism of Exo84c regulating stigma senescence. a, b** In wild type Arabidopsis, the senescence of stigmatic tissue induces autophagy that selectively degrades the exocyst-labelled structures (e.g. vesicles, MVB). This process requires the interaction between ER/EPCS localized VAP27 and Exo84c, and possibly the involvement of Exo70 subunits that interact with the ATG8 (**a**). In the *exo84c* mutant, the activity of autophagy is reduced in aged papilla cells, thus preventing cell death and delaying senescence (**b**).

### Yeast two-hybrid assays and yeast transformation

Full-length CDS of Exo84a, Exo84b and Exo84c were sub-cloned into the pGADT7-GW plasmid using Gateway LR reaction. VAP27-1 and VAP27-3 coding sequences without the transmembrane domains were also cloned into pGBKT7 plasmid by Gateway LR reaction. Both prey and bait plasmids were co-transformed into yeast AH109 strain and cultured on an SD/-Leu-Trp selective plate at 30 °C for 2–4 days. Three independent colonies of each combination were picked and dotted an SD/-Leu-His-Trp selection medium supplemented with 5 mM 3-AT. Combinations of pGBKT7-53 + pGADT7-T and pGBKT7-lam + pGADT7-T were used as positive and negative controls, respectively.

### Protein extraction and western blot analysis

Total proteins used for western blot analysis were extracted from 10-day-old Arabidopsis seedlings or *N. benthamiana* leaf segments. The tissue was ground in liquid nitrogen and mixed with an equal volume of 2 × SDS buffer containing 50 mM Tris-HCl, 40 mM NaCl, 10 mM MgCl$_2$, 5 mM EDTA and 5 mM DTT. The mixture was incubated at 95 °C for 10 min and then centrifuged at room temperature

to collect the supernatant. An appropriate amount of the total protein was loaded on 5% stacking gel, and separated on a 10% or 12% SDS-PAGE separating gel. After electrophoresis, the proteins were transferred to a nitrocellulose membrane. For detection, the membrane was incubated with primary antibody (1:500 for anti-VAP27; 1:300 for anti-Exo84c; 1:1000 for anti-Actin, BBI, D110007; 1:5000 for anti-HA-Tag, Yeasen, 30701ES60; 1:5000 for anti-Myc-Tag, Yeasen, 30601ES60; and 1:2000 for anti-GFP, Biorbyt orb323045) and a horseradish peroxidase (HRP)-conjugated goat anti-mouse/rabbit secondary antibody (Yeasen, 33201ES60, used at 1:5000; BBI, D110058, used at 1:5000) at room temperature. Please note original images of all immune-blot are available in the Source data document.

### Protein levels quantification analysis

The ratios of band intensities of target proteins to those of the loading controls (CBB or actin) were first calculated, and then the protein levels were calculated by being normalized to their corresponding mock controls.

## Co-IP assay

The co-immunoprecipitation assays were performed using GFP-Trap®_A (Chromotek, gtma-20) and followed the procedure as described previously[21,69]. In brief, approximately 0.5 g of 10-day-old Arabidopsis seedlings expressing Exo84cp::Exo84c:GFP or Exo84b-pro::Exo84b:GFP were used. The samples were ground in liquid nitrogen and resuspended in Nonidet P-40 buffer with 1 mM phenylmethylsulfonyl fluoride. The mixture was incubated for 1 h at 4 °C and then centrifuged at $2500 \times g$ for 2 min at 4 °C, the agarose pellet was harvested for western blot analysis.

## Confocal microscopy and image analysis

Images were collected using a laser scanning confocal microscope (Leica TCS SP8). Images were acquired with a 63× oil-immersion objective lens using the Leica PMT detector. The excitation/emission wavelengths for CFP, GFP, YFP and RFP/mCherry are 448 nm/460–500 nm, 488 nm/505–540 nm, 514 nm/550–590 nm and 552 nm/600–640 nm, respectively. FDA/PI staining of stigma papilla cells was performed as described by Gao et al.[5]. Images of whole stigmas were taken using a 20× dry objective lens. All images of each sample are representative of at least three independent replicates. The fluorescent intensity and band intensity (in western blot assay) were measured using Fiji.

## BiFC analysis

For BiFC analysis, full-length CDS of Exo84a, Exo84b and Exo84c were sub-cloned into pCL112-nYFP, and VAP27-1/-3 were sub-cloned into pCL113-cYFP plasmids, respectively. Images were taken with standard excitation/emission wavelengths for YFP.

## Chemical treatment

The chemical treatment was performed by transferring 5-day-old Arabidopsis seedlings or *N. benthamiana* leaf segments in liquid 1/2 MS medium containing Concanamycin A (1 μM) and/or Wortmannin (10 μM). The stigmatic papilla cells were treated by pasting lanolin containing 25 μM Conc A around the style as previously described[70,71]. Corresponding concentrations of DMSO were used as the controls in all experiments. For protein degradation rate analysis, 10-day-old Arabidopsis seedlings were treated in liquid 1/2 MS medium containing cycloheximide (50 μM).

## Pollination assay and silique clearing

Stage 12 flower buds were emasculated and pollinated with Col-0 pollen at 24, 48, and 72 HAE. The pistils were cut at the peduncle 4 h after pollination, then fixed for 2 h in ethanol-acetic acid (3:1), softened in 1 M NaOH at 60 °C for 1 h and stained with 0.01% (w/v) decolorized aniline blue for 2.5 h in 2% (w/v) $K_3PO_4$. Pistils were gently squashed onto a microscopic slide glass by placing the cover glass over the pistils. Samples were examined under a fluorescence microscope (Leica DMi8). For silique clearing, the siliques were harvested 10-14 days post-pollination and submerged in 75% ethanol for 3–5 days. Images were captured using a stereo fluorescence microscope (Leica M205 FA).

## Transmission electron microscopy

For chemical fixation of Arabidopsis roots, 5-mm root tips of 5-day-old Arabidopsis expressing UBQ10::GFP:Exo84c and VAP27-3p::VAP27-3:mCh following 1 μM Conc A treatment were collected and checked with confocal microscopy, then they were prefixed as in 2.5% glutaraldehyde (v/v in 0.1 M phosphate buffer, pH 7.2) for 2 h. Then the samples were rinsed with 0.1 M phosphate buffer for three times, 15 min each. The samples were post-fixed in 1% $OsO_4$ for another 2 h, and rinsed with 0.1 M phosphate buffer for three times as before. All the samples were dehydrated using an acetone gradient series of 30, 50, 70 and 90%, with a 20 min exposure to each acetone gradient and ending with the dehydration step in 100% acetone for three times (15 min each). The samples were then infiltrated in a graded scale of 3:1, 1:1, 1:3 (v/v) acetone/SPI-PON 812 resin and 100% SPI-PON 812 resin (SPI Supplies, West Chester, PA, USA) at the last step, each step was performed for 12 h. Samples were finally embedded in SPI-PON 812 resin and polymerized at 60 °C for 48 h. For transmission electron microscopy (TEM) analysis, the embedded samples were sectioned on an EM UC7 ultramicrotome (Leica, Germany). Ultrathin sections (80 nm thick) were stained with 2 % uranyl acetate and lead citrate and viewed using a TEM (Hitachi H-7650, Japan) at 80 kV.

## Statistical analyses

Statistical analyses were performed using the GraphPad Prism software (v.8.0). Statistical analyses of data were performed by two-tailed Student's $t$-test, one-way ANOVA with a post hoc Tukey's multiple comparisons test, two-way ANOVA with a post hoc Tukey's multiple comparisons test, and two-way ANOVA with a post hoc Dunnett's multiple comparisons test.

## Accession numbers

Accession numbers of genes in this article are VAP27-1 (AT3G60600), VAP27-3 (AT2G45140), Exo84a (AT1G10385), Exo84b (AT5G49830), Exo84c (AT1G10180), Sec6 (AT1G71820), Sec10 (AT5G12370) and Exo70B1 (AT5G58430).

## Reporting summary

Further information on research design is available in the Nature Portfolio Reporting Summary linked to this article.

# Data availability

The authors declare that all data supporting the findings of this study are available within the article and its Supplementary Information files, or from the corresponding author upon request. Source data are provided with this paper.

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

## Acknowledgements

We thank Prof. Zongcheng Lin (HZAU), Dr Patrick Duckney (Durham University) and Prof. Kunrong Mei (Tianjin University) for useful discussions about the manuscript. We thank Dr. Jianbo Cao and the Public Laboratory of Electron Microscopy, Huazhong Agricultural University. We thank Prof Yiqun Bao (Nanjing Agricultural University, China) for providing the transgenic Arabidopsis line expressing Sec6:GFP. We thank the Core Facilities at College of Life Science and Technology, Huazhong Agricultural University for assistance with microscopy. The project was supported by NSFC grants (32261160371, 92254307, 91854102), the Fundamental Research Funds for the Central Universities (2662023PY011) the Foundation of Hubei Hongshan Laboratory (2021hszd016) to P.W, and a China Postdoctoral Science Foundation (2021M691166) grant to T.Z.

## Author contributions

P.W. conceived and supervised the project. T.Z. performed most of the experiments. T.Z wrote and edited the manuscript with P.W. and P.J.H. Initial studies were performed in the lab of P.J.H. by J.Z. Y.L. helped with molecular cloning and transgenic plant generation. J.T.K. provided the *vap27 crispr* mutant. C.L. helped with TEM sample fixation and imaging. J.Z. X.Q. and E.G contribute to data interpretation and result discussion throughout the project.

## Competing interests

The authors declare no competing interests.
