## [Peer Review File · Nature Communications]

Exo84c interacts with VAP27 to regulate exocytotic compartment degradation and stigma senescenceReviewer #1 (Remarks to the Author):

The Manuscript by Zhang et al. reported an interesting interaction between Exo84c and VAP27-3 and a role of the interaction in selective autophagy of the exocyst complex during stigma senescence. The evidence of the interaction is solid and the data on the degradation of the exocyst complex in the vacuole is strong too. There are some points I think the authors should consider to revise/improve.

Major points:

(1) It appears that some GFP-Exo84a or Exo84b puncta were co-localized with VAP27-3-mCherry (yellow) (Figure S1b). The question is then, how specific is the interaction between Exo84c and VAP27-1 or -3? In Figure 1d, the expression of HA-cYFP-VAP27-3 and -1 with Exo84c were higher than those in -a and -b. Would this account for different BiFC signals revealed in Figure 1b-c?

(2) Co-expression of VAP27-3 relocated Exo84c to puncta as well as a reticular structure likely the ER (Figure S1b and 5b). Explain why the BiFC signal was only observed on puncta, not on the tubular ER.

(3) It would be nice if the authors also test the degradation of Exo84c and VAP27 in Figure 3e-f in the presence of MG132 (and perhaps ConCA as the control) to see/rule out if Exo84c and VAP27 are also degraded by the 26S complex.

(4) The biological significance of the degradation of the exocyst complex in stigma senescence. In line 537-540, the authors 'suggest that there may be ER stress response exists in ageing papilla cells. In this ER stress response, Exo84c acts as an adaptor to promote ER derived autophagosomes by interacting with VAP27, linking ER membrane to ATG8 ...' However, they also stated that '.. persistent ER stress will impact cell fate, by inducing autophagy, apoptosis or senescence.' There should be persistent ER stress in the mutants (likely, ER-phagy is defective as suggested by Figure 1c and 5c where ER puncta were made by the interaction and included in it), which would enhance stigma senescence? Alternatively, do the authors consider another possibility that some secretion activities (e.g. secretion of some cell wall enzymes) needed for cell senescence are mediated by Exocyst, such secretion may be inhibited in the mutants?

Minor points:

(1) Line 71-73, the format of GFP fused protein (Exo84c:GFP, Exo84a:GFP, Exo84b:GFP, Sec6:GFP) was not consistent throughout the paper, sometimes the format was Exo84b-GFP (line 505).

(2) Line 97, split-YFP assay should be BiFC.

Reviewer #2 (Remarks to the Author):

The authors of this manuscript report the isolation of Exo84c, a plant-specific Exo84, as a VAP27 interactor, and the observation that KO mutants of *exo84c* have a longer stigma lifetime, while OX of Exo84 and *vap27* mutations result in a shortening of the stigma lifetime. The authors propose that autophagy targets a group of molecules, including exocyst, which occurs associated with the ER during stigma cell death. While the proposed model is attractive, the presented data is insufficient to reach this conclusion. Additionally, Safavian et al. 2015 reported results that are the exact opposite of those presented in this manuscript, namely that the *exo84c* mutant has an abnormal stigma and poor seeding, and it is inappropriate that this report is not mentioned in this manuscript.

The data presented in Figure 1e do not allow for the determination of the specificity of the binding, because the amount of GFP-tagged Exo84 is totally different between Exo84b-GFP and GFP-Exo84c. It is also unclear why the location of GFP attached to Exo84 is reversed between Exo84b and Exo84c.

Regarding the immunoblot data, including Figure 3e, quantitative results are shown, but it is not stated whether these values are normalized using actin as an internal control or simply by measuring the intensity of the bands. Additionally, it is unclear whether the observed differences are truly meaningful, and the high background in Figure 3e makes it difficult to determine the credibility of the quantitative results.

Concerning the VAP27 antibody, while the paper by Wang et al. 2016 shows that it does not recognize VAP27-3, this manuscript claims that the antibody used recognizes VAP27-3 based on data in Fig. S3. Given that the vap27-3 mutation in c19 is 24 bp deletion, if this antibody recognizes VAP27-3, a protein with a small molecular weight would be detected.

Fig.5e, if comparing the degradation of CFP-ATG8e, the authors should also show the CFP-only band after degradation and compare the ratio of the full length to the degradation product. The same is for GFP-Exo84 in Fig.3g and 4j, and Sec6-GFP in Fig.6i.

It is unclear how the authors conclude that the structures shown in Fig. 5g, h, and 5j are ER-derived from the presented light microscopy and electron microscopy images. In particular, for the magnified electron micrographs, it is not clear whether these structures are truly autophagosome-derived. It is also unclear whether these images are magnifications of areas in the above photograph. Although it can be seen that these structures contain the ER, there is no evidence that the limiting membrane of these structures is derived from the ER membrane.

Line 476-477. "the activity of Exo84c is essential for this process" is an overstatement. Figure 6g shows that Sec6 is incorporated into the vacuole even in the exo84c mutant.

A more in-depth discussion is needed on how a mechanism that reduces reproduction efficiency can increase fitness.

In the complementation experiment with Exo84-GFP in Fig. S2, it is necessary to confirm whether Exo84a/b/c is expressed to the same extent by immunoblotting using the GFP antibody.

Reviewer #3 (Remarks to the Author):

In the manuscript "Exo84c regulates exocytotic compartment degradation and stigma senescence through a VAP27-mediated autophagy pathway" Zhang and colleagues propose a novel role for Exo84c and Vap27 in autophagy.

After demonstrating that Exo84c specifically interacts with Vap27, authors show that Exo84c negatively affects the lifetime of papillae cells, in contrast to Vap27, which has a positive role. They go on to show that Exo84c is transported into the vacuole by autophagy and propose that the interaction between Exo84c and Vap27 is required for this process. They also show reduced recruitment to the vacuole of Sec6 in absence of Exo84c, making the authors conclude that Exo84c may mediate the autophagy of the entire exocyst complex.

Although the topic of the study is very interesting and also relevant for potential applications, the main claims in the paper still require further experimental support. Notably, the title goes far beyond what was shown in this study.

Main points

1. One of the main discoveries of the study is what seems an interdependent degradation of Exo84c and Vap27, in other words authors propose that Vap27 and Exo84 affect each other's stability. However, their data does not fully support this. For figure 3e authors analyse Exo84 levels after 24h 50 μ M CHX treatment. CHX causes DNA damage and impairs critical cellular functions. Hence, CHX treatments are carried out for maximally for 12 h, at which point CHX may also not even be active, if not spiked in between. Hence, in a 24h treatment we may be rather measuring the reaction to toxicity and not degradation.

In addition, blots comparing protein levels between lines must be on the same membrane to be comparable (Figures 3e, 3f and 6i). Since showing an effect regarding the stability of the proteins is critical for the study, authors should show at least one with repetitions and statistical analysis. It

would also be important that the levels of the tested proteins be tested between lines (i.e. different genetic backgrounds as well as treatments).

Figure 3g is missing the controls. Proteins should be tested in the presence or absence of Vap27. What assay was used here for expression? Authors should provide more detailed information on the legends, which was usually quite scarce and also state how many biological repetitions were carried out.

Please also show total protein as a loading controls, and include molecular size marker in all blots.

2. Authors show that Vap27 coexpression with Exo84c leads to its recruitment to punctate compartments. This is a key observation and should be complemented with coexpression of the other two Exo84s, to whether its specific to Exo84c only. Of note, because there is such a clear relocalization to punctae, fluorescence intensity is an inadequate indicator for protein concentration, and should be excluded. The blot in Figure S5C should also be repeated to include the other Exo84s.

3. Another key observation in this study is the putative localization of Exo84c in punctae that colocalize with ER markers, which suggest that Exo84c is not acting as a canonical exocyst subunit and active in a process unrelated to secretion.

To determine the subcellular localization of the interaction between Exo84c and Vap27 authors use BiFC co-expressed transiently in *Nicotiana*. Because complementation of the split YFP is in principle irreversible, it leads to a constitutive interaction which is unphysiological and may results in artefactual localization. Hence, authors need to use a different approach to determine subcellular localization.

I also find it worrying that localization in punctae can only be achieved by co-overexpression of both proteins in tissues other than papillae cells, while when expressed on their own they show a different behaviour. This can be a problem of stoichiometry. However, proteins do not seem to behave similarly in the stable lines used in Figure 5G? Does also Sec6 localize to punctae when coexpressed with Vap27? And importantly, are such punctae also present in pistils?

Authors should provide evidence that the punctae are not artefacts of overexpression. This seems important, as it may lead to the conclusion that interaction takes place in the wrong site. As the exocyst itself is rather shown to be involved in tethering of post Golgi vesicles to the PM, it would make such data more even important. However, this could be reconciled by the observation that Vap27 was reported to link ER and PM (Wang et al. 2014 DOI 10.1016/j.cub.2014.05.003). Another question that poses itself within the same context, is whether Exo84c actually associates to the exocyst, as the authors propose that Exo84c recruits the entire complex. In the same vein, does Exo84c have a conserved CoRex?

4. Although authors show that the exo84c cannot be complemented by other isoforms, do Exo84a/b mutants show related phenotypes?

5. To further analyse the role of Exo84c-Vap27 interaction, authors coexpressed transiently both proteins in the presence of ATG8e. This experiment is problematic for two reasons. First, different number of constructs are used between experiments, which will affect transformation efficiency and by *Agrobacteria* an expression. The same problem holds true for Figure S9. Authors could include the other Exo84 isoforms as controls. It would also be interesting to determine whether the expression of ATG8 has an effect on the degradation. To further support their claims of increased autophagic flux, authors should show the cleaved GFP and determine the ratios.

6. Inhibition of vacuolar degradation was by ConA in lines coexpressing Vap27 and Exo84c resulted in the accumulation of both proteins autophagic body-like compartments, supporting a role of autophagy. This figure is quite interesting but it shows that both proteins clearly have a distinct localization, while Vap27 tendentially accumulates on the membrane, Exo84c maintains a punctate pattern. This could be indicative of the different role on a mechanistic level, which authors may want to include in the discussion.

7. Why do authors change from the Exo84c construct driven by the native promoter to the one driven by UBQ10? Does punctae formation require overexpression?

8. In the first sentence of the abstract authors claim that the exocyst plays a role in the regulation of autophagy. This is incorrect. Specific subunits of the exocyst have been shown to be degraded via autophagy, and in one example proposed to mediate the degradation of a protein. However, the exocyst has not been demonstrated to participate in the regulation of autophagy or participate in any way in the activation of autophagy, and is clearly down-stream in plants. The paper by Bodemann et al. (2011 DOI 10.1016/j.cell.2010.12.018.) shows a putative regulatory role, which however, remains controversial.

It is also not accurate to say that the exocyst regulates secretion, as it is a core component of the

secretory pathway. In general authors use the term "regulatory" out of its proper meaning.

9. In most tissues GFP is not fluorescent in the vacuole due to its low pH. However, authors seem to observe GFP fluorescence in the vacuole of papillae cells without treatment with ConA. If this is the case, it would suggest that vacuolar degradation in this type of cells is generally inactive, since vacuolar proteases require low pH.

10. Figure 1e is missing the equal loading control.

Minor points

- I missed in the introduction the function of the exocyst as a tethering complex and autophagy. Also, Vap27 is only briefly introduced in the results section, without mentioning for instance its role in autophagy (DOI: 10.1038/s41467-019-12782-6).

#51 What do authors mean by multiple cellular activities? Please elaborate.

#62 the *exo70B1* mutant has a spontaneous cell death phenotype, which is dependent on TN2. (Zhao et al. doi: 10.1371/journal.pgen.1004945). Its phenotypes are therefore, demonstratively not related to autophagy. The cited paper demonstrates that cell death phenotype is dependent on growth conditions, developmental stage and is activated before bolting and that plants are more susceptible to virulent *Pseudomonas*.

#67 *Exo84* function is rather unappreciated than underestimated. Consider changing

#151 Why do authors refer to Figure 2C which shows germination?

#172 At this point no regulatory mechanisms have been shown. Please rephrase

-Activity and concentration of a protein are two very different things. Please be more accurate.

- Authors refer frequently to the recruitment of components to the vacuole as "internalization".

However, internalization is most frequently used for endocytosis and could be confusing within the context of the vacuole, which in principle is also in the cell. Please consider changing.

-It is unclear what the point of Figure 5j is. What does it supposedly show? Please elaborate.

#75 What is a drastic autophagy? Please rephrase.

Reviewer #1 (Remarks to the Author):

The Manuscript by Zhang et al. reported an interesting interaction between Exo84c and VAP27-3 and a role of the interaction in selective autophagy of the exocyst complex during stigma senescence. The evidence of the interaction is solid and the data on the degradation of the exocyst complex in the vacuole is strong too. There are some points I think the authors should consider to revise/improve.

Major points:

(1) It appears that some GFP-Exo84a or Exo84b puncta were co-localized with VAP27-3-mCherry (yellow) (Figure S1b). The question is then, how specific is the interaction between Exo84c and VAP27-1 or -3? In Figure 1d, the expression of HA-cYFP-VAP27-3 and -1 with Exo84c were higher than those in -a and -b. Would this account for different BiFC signals revealed in Figure 1b-c?

Response 1: Thank you for your comments, we believe the co-localized signal between Exo84a/b and VAP27 is negligible. As the ER network links to multiple subcellular structures including autophagosomes, these yellow signals (in rare cases) may indicate random associations between these organelles. Please refer to a recently published paper on ER-autophagosome membrane interaction (Ye et al., PNAS, 2022, 119: e2205314119). In the revised manuscript, we have performed the Pearson coefficient analysis to further support our claim (Figure S1d of the revised manuscript).

With regard to Figure 1d, we are sorry about the confusion. Since the co-expression of VAP27 and Exo84c will promote their degradation, it is very difficult to see the band of GFP: Exo84c if the sample of Exo84c+VAP27 is equally loaded as Exo84a/b +VAP27 (please refer to Figure 3g of the original manuscript as an example). So, we have loaded more proteins for the immunoblot of Exo84c+VAP27 in order to see all protein bands clearly. We have clarified this in the figure legends. In general, together with the Co-IP, BiFC and Y2H data, we believe the interaction between Exo84 and VAP27 is specific to the Exo84c isoform.

(2) Co-expression of VAP27-3 relocated Exo84c to puncta as well as a reticular structure likely the ER (Figure S1b and 5b). Explain why the BiFC signal was only observed on puncta, not on the tubular ER.

Response 2: Thank you for your comment. This variation is likely caused by the fluorescence intensity of the VAP27-Exo84c labelled puncta, which is much stronger than their signal on the ER membrane. Therefore, in the BiFC experiments, the signals were mainly observed on puncta with the microscopy setting we applied. The ER background signal is detectable if a high detection setting is used, however, this would make the signal on the puncta over-saturated.

(3) It would be nice if the authors also test the degradation of Exo84c and VAP27 in Figure 3e-f in the presence of MG132 (and perhaps ConcA as the control) to see/rule out if Exo84c and VAP27 are also degraded by the 26S complex.

Response 3: We have performed this experiment as suggested by the reviewer, the result indicates that both VAP27 and Exo84c can be degraded through the 26s proteasomes as well. However, we feel this result is not so relevant to our study. There are many examples to demonstrate that an autophagy receptor/regulator can be ubiquitinated and degraded through 26s, regulating the activity of selective autophagy or being implicated in other pathways. For example, at the normal growth condition, autophagy

receptors can be ubiquitinated and degraded by the 26s complex, maintaining a low level of selective autophagy activity. Alternatively, autophagy receptor ubiquitination can also be a signal for autophagy activation and cargo recognition (please find more detail in Gubas et al., 2021). In plants, one of the examples is the ER-phagy receptor, RHD3, which can also be ubiquitinated by a E3 ligase Lunapark and subject to degradation through the 26s, regulating ER morphogenesis (Sun et al., 2020). Therefore, we prefer not to include this data, as it will not add much to our conclusion.

Figure R1. A cell-free immunoblot assay indicated that VAP27 (a) and Exo84c (b) can be degraded through the 26s proteasome pathway. Total proteins were extracted from the Col-0 plants with or without MG132.

(4) The biological significance of the degradation of the exocyst complex in stigma senescence. In line 537-540, the authors ‘suggest that there may be ER stress response exists in ageing papilla cells. In this ER stress response, Exo84c acts as an adaptor to promote ER derived autophagosomes by interacting with VAP27, linking ER membrane to ATG8 ...’ However, they also stated that ‘.. persistent ER stress will impact cell fate, by inducing autophagy, apoptosis or senescence.’ There should be persistent ER stress in the mutants (likely, ER-phagy is defective as suggested by Figure 1c and 5c where ER puncta were made by the interaction and included in it), which would enhance stigma senescence?

Alternatively, do the authors consider another possibility that some secretion activities (e.g. secretion of some cell wall enzymes) needed for cell senescence are mediated by Exocyst, such secretion may be inhibited in the mutants?

Response 4: Thank you for your suggestion, we have included these alternative interpretations in our revised manuscript (lines 577-581). Our preliminary results also indicated that ER-stress induced senescence is likely to be a common phenomenon in plants. Therefore, ER-stress could also be elevated at the ageing condition of the stigma tissue, this may activate the VAP27-Exo84c regulated autophagy pathway to induce cell death.

Minor points:

(1) Line 71-73, the format of GFP fused protein (Exo84c:GFP, Exo84a:GFP, Exo84b:GFP, Sec6:GFP) was not consistent throughout the paper, sometimes the format was Exo84b-GFP (line 505).

(2) Line 97, split-YFP assay should be BiFC.

Response 5: We have corrected these in the revised manuscript (line 103 of the revised manuscript), thank you for pointing out.

Reviewer #2 (Remarks to the Author):

The authors of this manuscript report the isolation of Exo84c, a plant-specific Exo84, as a VAP27 interactor, and the observation that KO mutants of *exo84c* have a longer stigma lifetime, while OX of Exo84 and *vap27* mutations result in a shortening of the stigma lifetime. The authors propose that autophagy targets a group of molecules, including exocyst, which occurs associated with the ER during stigma cell death. While the proposed model is attractive, the presented data is insufficient to reach this conclusion. Additionally, Safavian et al. 2015 reported results that are the exact opposite of those presented in this manuscript, namely that the *exo84c* mutant has an abnormal stigma and poor seeding, and it is inappropriate that this report is not mentioned in this manuscript.

Response 1: Thank you for your suggestion. We are sorry that the manuscript of Safavian et al. 2015 was missed in the first submission, we have added this reference in the revised manuscript. In this paper, they reported that the acceptance of compatible pollen was reduced in the *exo84c* mutant at the low humidity condition (below 35%), while at a high relative humidity condition (over 90%), the pollen acceptance defect can be partially restored. In our study, we are able to repeat their result of *exo84c* at a low humidity condition, but at the normal growth condition (with fluctuated humidity that is not monitored), pollen acceptance and plant reproduction is not affected using fresh stigma tissue (Fig 2a, g of the original manuscript). We have no idea why there is a slight difference, could be because of the variation in the plant growth environment. In the revised manuscript, we have discussed this briefly and provide our possible explanations (line 143-148 of the revised manuscript).

The data presented in Figure 1e do not allow for the determination of the specificity of the binding, because the amount of GFP-tagged Exo84 is totally different between Exo84b-GFP and GFP-Exo84c. It is also unclear why the location of GFP attached to Exo84 is reversed between Exo84b and Exo84c.

Response 2: Thank you for pointing it out. The expression of Exo84b is lower than Exo84c in the transgenic lines we have used, therefore, when the plant material was used equally, the band of Exo84b is weaker than Exo84c. In the revised manuscript, we have used different transgenic lines with similar expression levels, the result confirmed the interaction between VAP27 and Exo84c is specific.

With regard to the orientation of the GFP tag, we are sorry about the confusion. In the literature, Exo84b:GFP is commonly used, but N and C-terminal GFP are used interchangeably for other exocyst subunits. In our study, we mainly used two Exo84c constructs, UBQ10::GFP:Exo84c and Exo84cp::Exo84c:GFP, and found the position of GFP does not affect the protein behaviour of Exo84c, so we did not consider the effect of GFP position when performing our experiments. In the revised manuscript, we have used the *exo84c/Exo84cp::Exo84c:GFP* line instead to keep it consistent (Revised Figure 1e).

Regarding the immunoblot data, including Figure 3e, quantitative results are shown, but it is not stated whether these values are normalized using actin as an internal control or simply by measuring the intensity

of the bands. Additionally, it is unclear whether the observed differences are truly meaningful, and the high background in Figure 3e makes it difficult to determine the credibility of the quantitative results.

Response 3: These values are normalized using actin as an internal control, we have provided this information in the figure legends and the Methods section. To further confirm our results, we have repeated this experiment and quantified the band intensity in 3 independent assays, the result further indicates the degradation of Exo84c is reduced in two different *vap27-1/3* mutant lines. We don't feel the background signal will affect our result, as we only measure the signal of the band of Exo84c, the background signal was not included. We don't know why there is a background in the immune blot of endogenous Exo84c, but as such background signal is present in all tests, and its effects on the measurement are similar (can be treated as background noise). Please note we have provided a new set of images for Figure 3e, the time used for CHX treatment was reduced, as requested by Reviewer 3.

Concerning the VAP27 antibody, while the paper by Wang et al. 2016 shows that it does not recognize VAP27-3, this manuscript claims that the antibody used recognizes VAP27-3 based on data in Fig. S3. Given that the *vap27-3* mutation in c19 is 24 bp deletion, if this antibody recognizes VAP27-3, a protein with a small molecular weight would be detected.

Response 4: We thank this reviewer for pointing it out. We found the antibody information we provided in the original manuscript was not accurate, sorry about this error. The VAP27 antibody used here is freshly made in my lab using a similar protocol as reported before in Wang et al., 2016. We have revised this part of the method section. In the revised manuscript, we have further verified this antibody using *N. benthamiana* that transiently expresses VAP27-1:YFP or VAP27-3:YFP, both YFP (GFP) and VAP27 antibodies can detect the band of VAP27-1: YFP and VAP27-3:YFP at the same molecular weight.

With regard to the truncated VAP27-3 protein that might be generated in the CRISPR mutant, we have no idea why it cannot be recognized by our antibody. One possibility is the removal of this 24bp (8 a.a) from VAP27-3 may affect protein stability, so the level of this truncated protein is low and not detectable; alternatively, this 24bp deletion also overlaps with the sequence that we used as antigen, so the truncated protein may have less antigenicity and cannot be recognized by the antibody.

Fig.5e, if comparing the degradation of CFP-ATG8e, the authors should also show the CFP-only band after degradation and compare the ratio of the full length to the degradation product. The same is for GFP-Exo84 in Fig.3g and 4j, and Sec6-GFP in Fig.6i.

Response 4: As suggested, we have provided a new set of immunoblot of Figure 5e showing the free CFP band of CFP:ATG8e and Figure 6i showing the free GFP band of Sec6:GFP, and quantified the ratio of the full length to the degradation product. However, the quantification of free GFP is not possible for the Exo84-related constructs in original Figure 3g (Figure 3i of the revised manuscript) or Figure 4j, as the band of free GFP is very weak in these samples.

It is unclear how the authors conclude that the structures shown in Fig. 5g, h, and 5j are ER-derived from the presented light microscopy and electron microscopy images. In particular, for the magnified electron micrographs, it is not clear whether these structures are truly autophagosome-derived. It is also unclear whether these images are magnifications of areas in the above photograph. Although it can be seen that

these structures contain the ER, there is no evidence that the limiting membrane of these structures is derived from the ER membrane.

Response 5: In the revised manuscript, we have transformed a CFP:ATG8e marker into the GFP:Exo84c and VAP27-3:mCh Arabidopsis line, and repeat the study in Fig 5g-h (Fig. 5i-j in the revised manuscript). It clearly demonstrated that most of these Exo84c + VAP27 labelled puncta inside the vacuole after Conc A treatment are ER autophagosomes related, as indicated by the co-localization of ATG8 and VAP27 (an ER membrane protein, Fig. 5m of revised manuscript). Please note that VAP27 is solely ER and EPCS localized, so the structures labeled by VAP27 can be regarded as ER-derived without using an ER marker. Therefore, we believe the light microscopy data are sufficient to support the conclusion that ER-derived autophagosomes were formed.

On the other hand, we respectfully disagree with the comment "...no evidence that the limiting membrane of these structures is derived from the ER membrane...". As we know, autophagosome membranes are from multiple origins, not solely derived from the ER. ER-phagy is an autophagosome structure that degrades ER membranes, so the ER membrane could be regarded as cargoes here, not necessarily be the limiting membrane of the autophagosomes inside the vacuole. For TEM studies, we first checked our samples at the light microscopy level to make sure the Conc A treatment was successful (as seen in Fig 5j, revised manuscript), then fix the same sample for TEM. Therefore, we are certain that these structures accumulated in the vacuole (in Fig 5l, revised manuscript) were equivalent to the structures we have seen in Fig 5j (revised manuscript). In the revised manuscript, we have provided these details of TEM studies in the method section, and clearly indicated the origins of high magnification images, as we can see most of these structures contains ER membrane.

Line 476-477. "the activity of Exo84c is essential for this process" is an overstatement. Figure 6g shows that Sec6 is incorporated into the vacuole even in the exo84c mutant.

Response 6: We have rephrased it as "the activity of Exo84c is required for this process" (lines 523-524 of the revised manuscript).

A more in-depth discussion is needed on how a mechanism that reduces reproduction efficiency can increase fitness.

Response 7: The discussion section was revised. However, we do not understand the point raised by the reviewer that "reduces reproduction efficiency can increase fitness". We would like to hear further explanation if possible.

In the complementation experiment with Exo84-GFP in Fig. S2, it is necessary to confirm whether Exo84a/b/c is expressed to the same extent by immunoblotting using the GFP antibody.

Response 8: Thank you for pointing it out, we have performed the immunoblot assay as suggested (Figure S2e of the revised manuscript).

Reviewer #3 (Remarks to the Author):

In the manuscript "Exo84c regulates exocytotic compartment degradation and stigma senescence through a VAP27-mediated autophagy pathway" Zhang and colleagues propose a novel role for Exo84c and Vap27 in autophagy.

After demonstrating that Exo84c specifically interacts with Vap27, authors show that Exo84c negatively affects the lifetime of papillae cells, in contrast to Vap27, which has a positive role. They go on to show that Exo84c is transported into the vacuole by autophagy and propose that the interaction between Exo84c and Vap27 is required for this process. They also show reduced recruitment to the vacuole of Sec6 in absence of Exo84c, making the authors conclude that Exo84c may mediate the autophagy of the entire exocyst complex.

Although the topic of the study is very interesting and also relevant for potential applications, the main claims in the paper still require further experimental support. Notably, the title goes far beyond what was shown in this study.

Response 1: Thank you for your suggestions, we have revised the title to "Exo84c interacts with VAP27 to regulate exocytotic compartment degradation and stigma senescence". We would also like to hear any suggestions from the reviewer if possible.

Main points

1. One of the main discoveries of the study is what seems an interdependent degradation of Exo84c and Vap27, in other words authors propose that Vap27 and Exo84c affect each other's stability. However, their data does not fully support this. For figure 3e authors analyse Exo84c levels after 24h 50 μ M CHX treatment. CHX causes DNA damage and impairs critical cellular functions. Hence, CHX treatments are carried out for maximally for 12 h, at which point CHX may also not even be active, if not spiked in between. Hence, in a 24h treatment we may be rather measuring the reaction to toxicity and not degradation.

Response 2: In the revised manuscript, we have reduced the time for CHX treatment to 12 hours, and studied the stability of Exo84c and VAP27 in the different genetic backgrounds (Fig. 3e and 3g in the revised manuscript). The result further confirmed the interdependent degradation of Exo84c and VAP27.

In addition, blots comparing protein levels between lines must be on the same membrane to be comparable (Figures 3e, 3f and 6i). Since showing an effect regarding the stability of the proteins is critical for the study, authors should show at least one with repetitions and statistical analysis. It would also be important that the levels of the tested proteins be tested between lines (i.e. different genetic backgrounds as well as treatments).

Response 3: We have repeated the study as suggested and provided statistical analysis from 3 independent experiments. Please note, the original image of Figures 3e and 3g (revised manuscript) are from the same membrane (please see the figure below), it was cropped to fit the figure panel. Please note original images of all immune-blot are available in the Source data of supplementary documents that associated with this submission.

Figure R2. Original immunoblot of Figure 3e and 3g.

Figure 3g is missing the controls. Proteins should be tested in the presence or absence of Vap27.

Response 4: As suggested, we have used GFP:Exo84b as a control to prove that VAP27-3 specifically promote the degradation of Exo84c. GFP:Exo84b is singly expressed or co-expressed with VAP27-3:mCh in *N. benthamiana*, both the fluorescence intensity and the protein levels were detected and showed that the presence of VAP27-3:mCh would not affect the abundance of GFP:Exo84b (Fig. S6e-g in the revised manuscript).

In original Figure 3g (Fig. 3i of the revised manuscript), GFP:Exo84a and GFP:Exo84b can be regarded as the control of GFP:Exo84c, both of them were not degraded in the presence of VAP27-3:mCh. We believe this result is accurate because all of the protein combinations are infiltrated into the same *N. benthamiana* leaf with the same ODs.

What assay was used here for expression? Authors should provide more detailed information on the legends, which was usually quite scarce and also state how many biological repetitions were carried out. Please also show total protein as a loading controls, and include molecular size marker in all blots.

Response 5: We have provided this information in the figure legends, thank you for pointing it out. In most experiments, we used actin and coomassie brilliant blue gel (CBB) as the loading control or internal standards for protein level quantification.

2. Authors show that Vap27 coexpression with Exo84c leads to its recruitment to punctate compartments. This is a key observation and should be complemented with coexpression of the other two Exo84s, to whether its specific to Exo84c only. Of note, because there is such a clear relocalization to punctae, fluorescence intensity is an inadequate indicator for protein concentration and should be excluded. The blot in Figure S5C should also be repeated to include the other Exo84s.

Response 6: A similar study was performed in the original manuscript (Figure S1), where we showed VAP27-Exo84c co-localized at punctate structures. Please also refer to our response 1 to Reviewer 1. Also, we have quantified the protein concentration of Exo84b and Exo84c (+/- VAP27) using an immunoblot and quantified the fluorescence intensity as in Figure S6a-d, this result is shown in Figure S6e-g of the revised manuscript.

3. Another key observation in this study is the putative localization of Exo84c in punctae that colocalize with ER markers, which suggest that Exo84c is not acting as a canonical exocyst subunit and active in a process unrelated to secretion. To determine the subcellular localization of the interaction between Exo84c and Vap27 authors use BiFC co-expressed transiently in *Nicotiana*. Because complementation of the split YFP is in principle irreversible, it leads to a constitutive interaction which is unphysiological and may results in artefactual localization. Hence, authors need to use a different approach to determine subcellular localization.

Response 7: We have repeated this study by transiently co-expression of Exo84c, VAP27-3 and HDEL, the results are similar to our previous study using BiFC co-expressions. Please refer to Figure S1a of the revised manuscript.

I also find it worrying that localization in punctae can only be achieved by co-overexpression of both proteins in tissues other than papillae cells, while when expressed on their own they show a different behaviour. This can be a problem of stoichiometry. However, proteins do not seem to behave similarly in the stable lines used in Figure 5G? Does also Sec6 localize to punctae when coexpressed with Vap27? And importantly, are such punctae also present in pistils?

Authors should provide evidence that the punctae are not artefacts of overexpression. This seems important, as it may lead to the conclusion that interaction takes place in the wrong site. As the exocyst itself is rather shown to be involved in tethering of post Golgi vesicles to the PM, it would make such data more even important. However, this could be reconciled by the observation that Vap27 was reported to link ER and PM (Wang et al. 2014 DOI 10.1016/j.cub.2014.05.003).

Response 8: Thank you for your suggestions, we have repeated the same study using the stigma tissue from transgenic plants expressing GFP:Exo84c + VAP27-3:mCh (Figure 5c-d in the revised manuscript) or Sec6:GFP + VAP27-3:mCh (Figure S12 in the revised manuscript), the results also demonstrate a close association and co-localization between them. However, the VAP27-Exo84c labelled punctate structures were mainly observed during senescence, and were less abundant in fresh stigma tissue (Figure 5d in the revised manuscript), which further indicates that the VAP27-Exo84c labelled punctate structures are not artefacts but are senescence-induced in pistil tissue. Please note, plastids produce strong autofluorescence with the microscope settings we used here, but they can be easily distinguished from their morphology. In addition, such VAP27-Exo84c labelled puncta is also observed in the vacuole of papilla cells of Exo84cp:Exo84c-GFP/*exo84c* lines upon Conc A treatment (Figure S7a, revised manuscript), suggesting it behaves similarly regardless which promoter is used.

With regard to the site of VAP27-Exo84c interaction, we suggest it is likely initiated at the ER-PM contact sites. As the exocyst complex regulates the tethering of secretory vesicles at the PM, and VAP27 is a key player in mediating ER-PM interaction, it is reasonable to speculate that the interaction between Exo84c

and VAP27 drives a transient interaction between PM-tethered-vesicles and ER membrane, regulating the formation ER-phagy that initiates at the ER-PM interface.

Another question that poses itself within the same context, is whether Exo84c actually associates to the exocyst, as the authors propose that Exo84c recruits the entire complex. In the same vein, does Exo84c have a conserved CoRex?

Response 9: Thank you for your good suggestion. We have predicted the protein structures of Arabidopsis Exo84 isoforms using AlphaFold2 (Jumper et al., 2021; Varadi et al., 2021), the results show that compared to yeast Exo84, there are similar CoRex motifs in all three Arabidopsis Exo84 isoforms, indicating that Exo84c is likely associated to the other exocyst subunits.

In yeast, all eight subunits can be co-purified and stay associated at the same time from a structural biology perspective (Mei et al., 2018). With regard the association between Exo84c and other exocyst isoforms, we suggest it would be similar to the yeast homologue and the Exo84a/b isoforms.

Figure R3. The predicted structure of Exo84a, Exo84b and Exo84c, the CoRex motif is shown in blue.

4. Although authors show that the *exo84c* cannot be complemented by other isoforms, do Exo84a/b mutants show related phenotypes?

Response 10: Thank you for pointing it out. We have studied the phenotype of the *exo84a* mutant, which does not exhibit a stigma phenotype (Figure S3 in the revised manuscript). However, the *exo84b* homozygous mutant is lethal due to a severe defect in cell plate formation (Fendrych et al., 2010), so we cannot perform a similar analysis in this line.

5. To further analyse the role of Exo84c-Vap27 interaction, authors coexpressed transiently both proteins in the presence of ATG8e. This experiment is problematic for two reasons. First, different number of constructs are used between experiments, which will affect transformation efficiency and by Agrobacteria an expression. The same problem holds true for Figure S9. Authors could include the other Exo84 isoforms as controls. It would also be interesting to determine whether the expression of ATG8 has an effect on the degradation. To further support their claims of increased autophagic flux, authors should show the cleaved GFP and determine the ratios.

Response 11: We have repeated the experiments as suggested. In control experiments, Exo84b, VAP27-3 and ATG8 are co-expressed (Figure S9f), and the result suggested that no degradation was found (as indicated by the ratio of Free CFP/full-length protein). Only the expression of Exo84c and VAP27-3 is able to enhance autophagic flux, as indicated by the reduction of the full-length CFP:ATG8 and the increase of free CFP band (Figure 5g). These data are provided in the revised manuscript.

In addition, we have also co-expressed another Exo84 isoform with VAP27-3:YFP and Exo70B1:mCh/Sec10:mCh in *N. benthamiana* (Figure S11 in revised manuscript), and no colocalization is observed in contrast to Figure S10.

6. Inhibition of vacuolar degradation was by ConA in lines coexpressing Vap27 and Exo84c resulted in the accumulation of both proteins autophagic body-like compartments, supporting a role of autophagy. This figure is quite interesting but it shows that both proteins clearly have a distinct localization, while Vap27 tendentially accumulates on the membrane, Exo84c maintains a punctate pattern. This could be indicative of the different role on a mechanistic level, which authors may want to include in the discussion.

Response 12: Thank you for pointing it out. We have revised the discussion, and provide some in-depth interpretations on the mechanism of VAP27-regulated autophagy (Line 560-564). In a recent paper from Liwen Jiang's lab, they suggested that VAP27 is part of the autophagy machinery that links the ER membrane and autophagosomes. In our case, VAP27 may also act as an adaptor that recruits the Exocytic compartment and the core autophagy complex, facilitating the degradation of the exocytotic compartment.

7. Why do authors change from the Exo84c construct driven by the native promoter to the one driven by UBQ10? Does punctae formation require overexpression?

Response 13: Please refer to our response 8 above, VAP27-Exo84c labelled puncta is also observed in Exo84cp:Exo84c-GFP/*exo84c* lines, indicating their localization on autophagosome-like structures do not require over-expression. We use the UBQ10:Exo84c-GFP construct because it gives a better signal-to-noise ratio, this is extremely important for live cell imaging in papillae cells, as the auto-fluorescence of plastids is always there, and it could be more pronounced with a high-power laser that used to detect the weak GFP signal. However, to exclude the unlikely artefact of the UBQ10 promoter, we have confirmed the vacuole accumulation of Exo84c:GFP in both root and papillae cells of the Exo84cp:Exo84c-GFP/*exo84c* lines (Figure S8a and d). The results are similar.

8. In the first sentence of the abstract authors claim that the exocyst plays a role in the regulation of autophagy. This is incorrect. Specific subunits of the exocyst have been shown to be degraded via autophagy, and in one example proposed to mediate the degradation of a protein. However, the exocyst has not been demonstrated to participate in the regulation of autophagy or participate in any way in the activation of autophagy, and is clearly down-stream in plants. The paper by Bodemann et al. (2011 DOI 10.1016/j.cell.2010.12.018.) shows a putative regulatory role, which however, remains controversial. It is also not accurate to say that the exocyst regulates secretion, as it is a core component of the secretory pathway. In general authors use the term "regulatory" out of its proper meaning.

Response 14: Thank you for pointing it out, and we agree with the reviewer. We have rephrased the abstract and remove the term “regulate”.

9. In most tissues GFP is not fluorescent in the vacuole due to its low pH. However, authors seem to observe GFP fluorescence in the vacuole of papillae cells without treatment with ConA. If this is the case, it would suggest that vacuolar degradation in this type of cells is generally inactive, since vacuolar proteases require low pH.

Response 15: Yes, the accumulation of the Exo84c-VAP27 signal is found in the vacuole of ageing papillae cells (even without Conc A treatment), while their accumulation can be further enhanced by an additional Conc A treatment. This effect is very different to cells from the vegetative tissue. At this stage, we are not clear why this happens. One possible explanation is that the cytoplasm-to-vacuole transport in papilla cells is very active during pollen-pistil recognition (pollen on pistil germination occurs within 15 min, so a lot of signalling and protein trafficking events need to take place rapidly). As a result, some Exo84/VAP27 signal accumulation in the vacuole is observed as the vacuole degradation activity might be saturated. Alternatively, the pH in papilla cells may increase during senescence after 24 HAE, so the GFP signal was accumulated gradually. However, we don't have solid evidence for this hypothesis at the moment, and we believe it is out of the scope of our study.

10. Figure 1e is missing the equal loading control.

Response 16: We have added the loading control as suggested.

Minor points

- I missed in the introduction the function of the exocyst as a tethering complex and autophagy. Also, Vap27 is only briefly introduced in the results section, without mentioning for instance its role in autophagy (DOI: 10.1038/s41467-019-12782-6). #51 What do authors mean by multiple cellular activities? Please elaborate.

Response 17: We have revised it as suggested, please refer to line 55-60.

#62 the exo70B1 mutant has a spontaneous cell death phenotype, which is dependent on TN2. (Zhao et al. doi: 10.1371/journal.pgen.1004945). Its phenotypes are therefore, demonstratively not related to autophagy. The cited paper demonstrates that cell death phenotype is dependent on growth conditions, developmental stage and is activated before bolting and that plants are more susceptible to virulent Pseudomonas.

Response 18: Thank you for pointing it out, we have deleted the introduction of exo70B1 mutant.

#67 Exo84 function is rather unappreciated than underestimated. Consider changing

#151 Why do authors refer to Figure 2C which shows germination?

#172 At this point no regulatory mechanisms have been shown. Please rephrase

-Activity and concentration of a protein are two very different things. Please be more accurate.

- Authors refer frequently to the recruitment of components to the vacuole as “internalization”. However, internalization is most frequently used for endocytosis and could be confusing within the context of the vacuole, which in principle is also in the cell. Please consider changing.

-It is unclear what the point of Figure 5j is. What does it supposedly show? Please elaborate.

Response 19: We have rephrased the statements as the reviewer suggested. The TEM studies aim to show that the structures accumulated in the vacuole (in Fig 5i, revised manuscript) were very likely the structures we have seen in Fig 5j at the ultrastructural level (revised manuscript). And these structures are ER-derived.

#75 What is a drastic autophagy? Please rephrase.

Response 20: We have rephrased the statements as the reviewer suggested.

Reference

Ye H, Gao J, Liang Z, Lin Y, Yu Q, Huang S, Jiang L. *Arabidopsis* ORP2A mediates ER-autophagosomal membrane contact sites and regulates PI3P in plant autophagy. *Proc Natl Acad Sci U S A*. 2022 Oct 25;119(43):e2205314119. doi: 10.1073/pnas.2205314119. Epub 2022 Oct 17. PMID: 36252028; PMCID: PMC9618059.

Gubas A, Dikic I. A guide to the regulation of selective autophagy receptors. *FEBS J*. 2022 Jan;289(1):75-89. doi: 10.1111/febs.15824. Epub 2021 Apr 5. PMID: 33730405.

Sun J, Movahed N, Zheng H. LUNAPARK Is an E3 Ligase That Mediates Degradation of ROOT HAIR DEFECTIVE3 to Maintain a Tubular ER Network in Arabidopsis. *Plant Cell*. 2020 Sep;32(9):2964-2978. doi: 10.1105/tpc.18.00937. Epub 2020 Jul 2. PMID: 32616662; PMCID: PMC7474291.

Jumper J, Evans R, Pritzel A, Green T, Figurnov M, Ronneberger O, Tunyasuvunakool K, Bates R, Žídek A, Potapenko A, Bridgland A, Meyer C, Kohl SAA, Ballard AJ, Cowie A, Romera-Paredes B, Nikolov S, Jain R, Adler J, Back T, Petersen S, Reiman D, Clancy E, Zielinski M, Steinegger M, Pacholska M, Berghammer T, Bodensteiner S, Silver D, Vinyals O, Senior AW, Kavukcuoglu K, Kohli P, Hassabis D. Highly accurate protein structure prediction with AlphaFold. *Nature*. 2021 Aug;596(7873):583-589. doi: 10.1038/s41586-021-03819-2. Epub 2021 Jul 15. PMID: 34265844; PMCID: PMC8371605.

Varadi M, Anyango S, Deshpande M, Nair S, Natassia C, Yordanova G, Yuan D, Stroe O, Wood G, Laydon A, Žídek A, Green T, Tunyasuvunakool K, Petersen S, Jumper J, Clancy E, Green R, Vora A, Lutfi M, Figurnov M, Cowie A, Hobbs N, Kohli P, Kleywegt G, Birney E, Hassabis D, Velankar S. AlphaFold Protein Structure Database: massively expanding the structural coverage of protein-sequence space with high-accuracy models. *Nucleic Acids Res*. 2022 Jan 7;50(D1):D439-D444. doi: 10.1093/nar/gkab1061. PMID: 34791371; PMCID: PMC8728224.

Mei K, Li Y, Wang S, Shao G, Wang J, Ding Y, Luo G, Yue P, Liu JJ, Wang X, Dong MQ, Wang HW, Guo W. Cryo-EM structure of the exocyst complex. *Nat Struct Mol Biol.* 2018 Feb;25(2):139-146. doi: 10.1038/s41594-017-0016-2. Epub 2018 Jan 15. Erratum in: *Nat Struct Mol Biol.* 2018 Dec;25(12):1137. PMID: 29335562; PMCID: PMC5971111.

Fendrych M, Synek L, Pecenková T, Toupalová H, Cole R, Drdová E, Nebesárová J, Sedinová M, Hála M, Fowler JE, Zársky V. The Arabidopsis exocyst complex is involved in cytokinesis and cell plate maturation. *Plant Cell.* 2010 Sep;22(9):3053-65. doi: 10.1105/tpc.110.074351. Epub 2010 Sep 24. PMID: 20870962; PMCID: PMC2965533.

Reviewer #1 (Remarks to the Author):

I think authors have addressed all my comments. The only suggestion I have is to include that result that both VAP27 and Exo84c are also degraded through the 26s proteasomes in the paper. I believe that it is good to let readers know that both proteins are also degraded in another degradation system. The potential scientific significance of this 26S based degradation can also be discussed like what they write in the rebuttal letter. It will not relegate, but enhance the significance of the manuscript.

Reviewer #2 (Remarks to the Author):

In the revised manuscript, the authors addressed concerns that I raised in the previous round of reviewing. Transgenic plants and reagents reported in this manuscript would be useful for researchers working in related fields.

Reviewer #1 (Remarks to the Author):

I think authors have addressed all my comments. The only suggestion I have is to include that result that both VAP27 and Exo84c are also degraded through the 26s proteasomes in the paper. I believe that it is good to let readers know that both proteins are also degraded in another degradation system. The potential scientific significance of this 26S based degradation can also be discussed like what they write in the rebuttal letter. It will not relegate, but enhance the significance of the manuscript.

> We are thankful for the reviewer's suggestion. However, we still feel that the proteasomal degradation results of Exo84c and VAP27 are not closely related to our study. We mainly focused on the effects of autophagic degradation of Exo84c-VAP27-mediated exocyst compartments during stigma senescence in this study. And the live-cell imaging of colocalization of VAP27 and Exo84c/exocyst subunits during the senescence of stigmatic papilla supported our conclusions well. In contrast, besides the degradation assay we have provided in the first round of review, we have no more evidence to indicate the role of proteasomal degradation of Exo84c and VAP27 plays in stigma ageing process. So we think it is better not to include the 26S degradation data into our final manuscript.

Reviewer #2 (Remarks to the Author):

In the revised manuscript, the authors addressed concerns that I raised in the previous round of reviewing. Transgenic plants and reagents reported in this manuscript would be useful for researchers working in related fields.

> We are grateful for the reviewer's constructive suggestions which make our manuscript better for publication.